# VERT: A SystemVErilog Assertion Dataset to improve HaRdware VerificaTion with LLMs

## Abstract

Hardware verification is a critical step in the modern System-on-Chip (SoC) design cycle, consuming approximately 70% of development time. SystemVerilog assertions are pivotal in the verification process, ensuring that designs function as intended. However, existing industrial practices rely on manual assertion generation, which becomes increasingly untenable as hardware systems become complex. Recent research has explored the potential of Large Language Models (LLMs) to automate the hardware verification process, reducing human intervention. Despite this, State-of-the-Art (SOTA) proprietary models, such as OpenAI's GPT-4o, have shown limitations in generating accurate assertions and require costly licenses and restricted usage. While smaller, open-source LLMs offer a more accessible option, they require fine-tuning to handle the complexities of the source code and generate accurate assertions. This highlights the need for a dataset that enables these models to achieve superior performance compared to SOTA LLMs. To this end, we present VERT, a dataset designed to improve the generation of SystemVerilog assertions using LLMs. Our dataset empowers researchers and hardware corporations to fine-tune smaller, open-source LLMs, surpassing larger proprietary models such as GPT-4 in accuracy and efficiency. Furthermore, VERT eliminates the need for expensive licenses and ensures data privacy through local fine-tuning, providing a scalable, cost-effective solution for automated hardware verification. To curate the dataset, we systematically compile and augment variables from open-source hardware description languages (HDL), generating conditions to create synthetic code snippets paired with corresponding assertions. We show that smaller, open-source LLMs, such as Deepseek Coder 6.7B and Llama 3.1 8B, when fine-tuned on VERT, outperform OpenAI's GPT-4o in assertion generation. The assertions generated by the fine-tuned models are evaluated on industry-standard platforms, including OpenTitan, CVA6, and Pulpissimo SoCs, demonstrating up to a 96.88% improvement in both functional and syntactical correctness compared to the base models and up to 24.14% when compared to GPT-4o. This demonstrates the prowess of VERT in enabling researchers to potentially reduce the overhead and human error associated with manual assertion generation, offering a scalable solution for industry-grade hardware designs. The dataset is available at https://anonymous.4open.science/r/VERT-4D6D/.

## 1 Introduction

In modern computing, System-on-Chip (SoC) designs have become dominant, offering extensive integration of various Intellectual Property (IP) cores into a single chip Miftah et al. (2024). While this approach significantly reduces production timelines and lowers costs, it also introduces critical challenges. One of the most pressing issues is the detection of functional bugs in these complex designs, which can consume up to 70% of the overall development time Farahmandi et al. (2020). Failure to detect design bugs prior to chip fabrication can lead to significantly higher post-production costs. This emphasizes the necessity of rigorous pre-manufacturing verification processes to identify and resolve potential issues early. Early detection mitigates the need for costly redesigns and minimizes production delays. Consequently, thorough hardware verification before fabrication is essential to ensure the design operates as intended and meets performance requirements.

Hardware assertions play a crucial role in addressing these verification needs. Typically expressed through SystemVerilog Assertions (SVA), they enable early bug detection by capturing critical system properties. However, manually generating these assertions is a time-consuming process that relies heavily on designers' expertise, making it challenging to adapt to complex designs and prone to human error Dessouky et al. (2019); Fang et al. (2024).

Large Language Models (LLMs) offer a solution by utilizing insights gleaned from textual data, such as code, to address these limitations by automatically generating SVA. This process automates the otherwise tedious manual task of assertion writing, ensuring a significant reduction in time and human effort and leading to more efficient verification cycles. However, recent academic research shows that proprietary and open-source LLMs struggle with generating high-quality Verilog code, including assertions. Even models such as Open AI's GPT-4 perform poorly in Verilog code generation due to a lack of high-quality, model-specific tuning data Zhao et al. (2024). This is further substantiated by recent research, which showed that only 11% of the SVAs generated by GPT-4 on the OpenTitan SoC were unique and correct Kande et al. (2024).

Specifically, these LLMs often generate SVAs that are neither syntactically nor functionally correct, necessitating human intervention. These issues are discussed in detail in Section 3. Conversely, employing a Verilog code dataset tailored to hardware design can significantly enhance the generation capabilities of LLMs Liu et al. (2024). Thus, curating open-source, high-quality, hardware-specific datasets is crucial for utilizing LLMs to their full potential in hardware design and verification.

To this end, we introduce VERT, a large-scale, high-quality, open-source dataset explicitly designed for formal and dynamic verification. Our dataset addresses the limitations of proprietary models such as OpenAI's GPT-4o, which require costly licenses and restrict usage. By empowering researchers and hardware corporations to fine-tune smaller, open-source LLMs, we aim to enable models that can outperform GPT-4o in generating SystemVerilog assertions. The key advantage of our dataset is that it allows smaller, more efficient models to achieve higher accuracy and functionality than larger, licensed LLMs without the associated costs or restrictions. By open-sourcing VERT, we not only enable local fine-tuning to safeguard sensitive design data but also provide a solution that enhances both performance and accessibility. Our ultimate goal is to demonstrate that with the right dataset, even compact, open-source models can deliver superior results, offering a cost-effective and scalable foundation for automated hardware verification.

Our work introduces several key contributions to hardware verification using LLMs,

- We introduce VERT, an open-source dataset specifically designed for SystemVerilog assertion generation. This dataset addresses the limitations of existing proprietary models and provides a valuable resource for advancing research in hardware verification.

- VERT addresses the challenges faced by LLMs in assertion generation by providing a carefully curated set of diverse cases–including standardized clock cycle interpretations, deeply nested conditions, and long logical expressions, thereby enhancing its ability to generate logically consistent assertions without oversimplifying or omitting critical conditions.

- VERT enables smaller, open-source LLMs such as DeepSeek Coder 6.7B and Llama 3.1 8B to surpass the performance of order of magnitude larger proprietary models like GPT-4o by up to 24.14% in generating accurate assertions.

- We perform an extensive evaluation of different LLMs fine-tuned with VERT on metrics such as syntactical and functional correctness. In specific, the models can achieve up to 100% on both the correctness measure while tested on modules from industry-standard SoCs, including OpenTitan, Pulpissimo, and CVA6, demonstrating the reliability and effectiveness of VERT to be used for real-world fine-tuning.

## 2 BACKGROUND AND RELATED WORKS

### 2.1 HARDWARE VERIFICATION

As modern hardware designs grow in complexity, ensuring their functional correctness has become increasingly challenging Ziegler et al. (2017). Hardware verification plays a critical role in guaranteeing that these designs meet their specifications and remain error-free Gupta (1992) Two major

approaches are commonly used in hardware verification: formal and dynamic (or simulation-based) verification. Each approach leverages either a golden reference model (GRM) or assertions. GRMs are typically restricted to dynamic verification, which simulates hardware behavior to check against expected outcomes. However, assertions offer greater flexibility, as they can be applied in formal and dynamic/simulation-based verification environments Miftah et al. (2024); Zhang et al. (2018).

Assertions in formal verification mathematically prove whether design properties can be violated, ensuring critical behaviors are maintained. In dynamic verification, assertions monitor execution and flag violations, helping identify errors early and reducing the risk of critical failures. Despite their importance, assertions are traditionally manually written by designers or verification engineers. This manual process is both time-consuming and prone to human error, especially in large, complex systems. The limited scalability of manually generated assertions contributes to longer development cycles and increases the risk of incomplete verification coverage, highlighting the need for automation in this domain.

## 2.2 LLMs for hardware design

LLMs are a groundbreaking development in artificial intelligence, leveraging vast datasets to generate human-like text with remarkable accuracy. Popular models such as OpenAI's GPT series have demonstrated exceptional performance in tasks like language translation, summarization, and sentiment analysis, thanks to their contextual understanding enabled by attention mechanisms Roumeliotis & Tselikas (2023); Devlin et al. (2018). Existing research has explored the use of LLMs for hardware design and verification Krishnamurthy & Hsiao (2020); Aditi & Hsiao (2023); Wan et al. (2024); Fang et al. (2024); Kande et al. (2024); Zhang et al. (2023); Orenes-Vera et al. (2023); Tarek et al. (2024a); Srivastava et al. (2023); Qayyum et al. (2024); Tarek et al. (2024b); Blocklove et al. (2024). However, as noted by researchers, such models often make errors as they lack knowledge of HDL languages and hardware assertion. A study by Liu et al. (2024) showed that this problem can be solved by creating datasets that can provide LLMs with such knowledge. To this end, researchers have generated Verilog datasets Thakur et al. (2023); Zhang et al. (2024). However, these datasets primarily focus on hardware design and do not explicitly target assertion generation. VERT is intended to fill this gap by specifically addressing hardware verification assertions, providing a more targeted resource for training LLMs to understand and generate assertion-based verification code.

## 3 Motivation

This section discusses challenges faced by state-of-the-art (SoTA) LLMs, like GPT-4o, in generating SystemVerilog assertions for hardware verification. This motivates the need for a carefully curated dataset designed to improve the ability of the model to generate precise and reliable assertions.

**Clock Cycle Misinterpretations:** A common issue with models is their inability to capture clock-cycle delays in assignment logic accurately. LLMs often generate assertions referencing incorrect or future clock cycles, leading to faulty hardware verification. As illustrated in Textbox 3.1, the source code is sensitive to the clock's rising edge and the reset signal's falling edge. When a reset signal is received, assignments should occur within the same clock cycle. However, the generated assertions use the "| =>" symbol, indicating that the assignment should be verified in the next clock cycle. This suggests a model bias towards checking assignments in the cycle following the satisfaction of the precondition. Therefore, it is necessary to fine-tune the model to enable accurate capturing of the clock cycle delay from the source code. This can be inferred from the 'always'-block type and the sensitivity list. For example, the 'always_comb' block means everything under this block is combinatorial and, therefore, instantaneous.

**Miscapturing *if* Condition for *else* Branches:** LLMs often overlook the previous *if/else if* conditions when generating assertions for the subsequent *else* or *else if* branches in conditional statements. For instance, in Textbox 3.2, the second assertion should verify that 'monitor_enable' is FALSE and 'monitor_enable_q' is TRUE. Next, if 'bus_release_cnt_load' is TRUE, the subsequent assignment check should occur.

---

3.1: Clock Cycle and Pre-condition Capturing in GPT-4o Outputs.

Generate a list of synchronous SystemVerilog assertion executing at (*posedge* clk_i) from the following code:

```
1  always @(posedge clk_i or negedge rst_ni) begin
2      if (~rst_ni) begin
3          wr_state_q <= IDLE;
4          wr_cnt_q   <= '0;
5      end else begin
6          wr_state_q <= wr_state_d;
7          wr_cnt_q   <= wr_cnt_d;
8      end
9  end
```

Response:

```
1  ...
2  // Assertion to ensure that wr_state_q follows wr_state_d when rst_ni is high
3  assert property (@(posedge clk_i) (rst_ni && (wr_state_d !== IDLE)) |=> (wr_state_q ==
        wr_state_d))
4    else $error("State transition failure: wr_state_q did not follow wr_state_d.");
5
6  // Assertion to ensure that wr_cnt_q follows wr_cnt_d when rst_ni is high
7  assert property (@(posedge clk_i) (rst_ni && (wr_cnt_d !== '0)) |=> (wr_cnt_q == wr_cnt_d))
8    else $error("Counter transition failure: wr_cnt_q did not follow wr_cnt_d.");
9
10  // Assertion to check that the state and count do not update during reset
11  assert property (@(posedge clk_i) (~rst_ni) |=> (wr_state_q == IDLE && wr_cnt_q == '0))
12    else $error("Reset condition not maintained: wr_state_q or wr_cnt_q changed during reset.");
13  ...
```

**Nested *If-Else* Conditions:** LLMs often struggle with generating assertions for deeply nested if-else conditions (i.e., blocks nested beyond two levels), which are common in hardware design. In Textbox 3.2, GPT-4o incorrectly represents the logic of a nested block. Specifically, it ignores the *if* condition and creates an assertion using a ternary operator, where the inner *if* condition is used as the test, the assignment as the true branch, and 1'b1 for the false branch. This assertion has two issues: first, SVA does not support ternary operators; second, the original code lacks an *else* condition, meaning the value should remain unchanged when the condition is false.

---

3.2: Relation Representation for Nested Conditional Blocks in GPT-4o Outputs.

Generate a list of asynchronous SystemVerilog Assertion from the following code

```
1  if (!rst_ni) begin
2      bus_release_cnt <= '0;
3  end else if (monitor_enable &&!monitor_enable_q) begin
4      if (multi_controller_enable_i) begin
5          bus_release_cnt <= 30'(bus_inactive_timeout_i);
6      end
7  end else if (bus_release_cnt_load) begin
8      bus_release_cnt <= bus_release_cnt_sel;
9  end else if (bus_release_cnt_dec && (bus_release_cnt!= '0)) begin
10      bus_release_cnt <= bus_release_cnt - 1'b1;
11  end
```

Response

```
1  ...
2  // Synchronous assertion for monitor enable signal transition
3  assert property (@(posedge clk) disable iff (!rst_ni)
4    (monitor_enable && !monitor_enable_q) |-> (multi_controller_enable_i ?
5    (bus_release_cnt == bus_inactive_timeout_i) : 1'b1)
6  ) else $fatal("Bus release count should be set to bus_inactive_timeout_i when monitor_enable
        rises and multi_controller_enable_i is high.");
7
8  // Synchronous assertion for bus_release_cnt_load condition
9  assert property (@(posedge clk) disable iff (!rst_ni)
10    (bus_release_cnt_load) |-> (bus_release_cnt == bus_release_cnt_sel)
11  ) else $fatal("Bus release count should be loaded with bus_release_cnt_sel when
        bus_release_cnt_load is asserted.");
12  ...
```

**Handling Long Conditions:** Another challenge with LLMs is their difficulty capturing long conditions involving multiple logical operators and variables. These complex conditions often result in overly simplistic or incorrect assertions. For instance, in Textbox 3.3, the source code requires any

one of six conditions to be met before raising the 'mr_err' flag. However, the generated output creates three assertions for one flag change, neglecting the other three conditions. This not only leads to redundant assertions but also causes overhead in the number of assertions. Furthermore, LLMs may mix assignment checks with preconditions, producing incorrect assertions. The assertions provided in the response in Textbox 3.3 are incorrect because they are structured with a logical flaw in how the conditional checks and error-handling mechanisms are designed. Specifically, the assertions only trigger an error when both the condition (e.g., `key_full_sel_o != mr_key_full_sel[i]`) and the error flag (`mr_err == 1'b1`) are true. This leads to an issue where the check fails to capture all error conditions properly. Ideally, the assertion should report an error when any of the conditions provided fails.

---

**3.3: Handling Long Conditions in GPT-4o Outputs.**

Generate a list of asynchronous systemverilog assertion from the following code

```
1    if (state_sel_o       != mr_state_sel[i]     ||
2        add_rk_sel_o      != mr_add_rk_sel[i]    ||
3        key_full_sel_o   != mr_key_full_sel[i]  ||
4        key_dec_sel_o    != mr_key_dec_sel[i]   ||
5        key_words_sel_o  != mr_key_words_sel[i] ||
6        round_key_sel_o  != mr_round_key_sel[i]) begin
7      mr_err = 1'b1;
8    end
```

Response

```
1  ...
2  // Asynchronous assertion to check key_full_sel_o against mr_key_full_sel[i]
3  assert property ((key_full_sel_o != mr_key_full_sel[i]) |-> (mr_err == 1'b1))
4    else $error("Asynchronous check failure: key_full_sel_o does not match mr_key_full_sel[%0d]
        and mr_err is not asserted.", i);
5
6  // Asynchronous assertion to check key_dec_sel_o against mr_key_dec_sel[i]
7  assert property ((key_dec_sel_o != mr_key_dec_sel[i]) |-> (mr_err == 1'b1))
8    else $error("Asynchronous check failure: key_dec_sel_o does not match mr_key_dec_sel[%0d] and
        mr_err is not asserted.", i);
9
10 // Asynchronous assertion to check key_words_sel_o against mr_key_words_sel[i]
11 assert property ((key_words_sel_o != mr_key_words_sel[i]) |-> (mr_err == 1'b1))
12   else $error("Asynchronous check failure: key_words_sel_o does not match mr_key_words_sel[%0d]
        and mr_err is not asserted.", i);
13 ...
```

---

Therefore, addressing these challenges through a structured dataset is essential to improving the LLM's ability to generate accurate SystemVerilog assertions, ensuring better alignment with the underlying hardware design logic. Examples of our approach to addressing these challenges are presented in the Appendix A.5.1.

## 4 PROPOSED VERT DATASET

Our proposed dataset VERT, is a curated collection of Verilog/SystemVerilog code snippets paired with SystemVerilog assertions, designed to fine-tune LLMs for generating syntactically and functionally correct assertions for hardware verification. By addressing the biases and errors with existing LLM-generated assertions (as mentioned in Section 3), we aim to improve the LLM's handling of complex SystemVerilog assertions and enhance the overall reliability of its outputs.

### 4.1 INTUITION IN DATASET FORMULATION

**Clock Cycle Misinterpretations:** To resolve clock cycle misinterpretation, we standardized our format by using the overlapping implication symbol $(|->)$ with a specified delay count, replacing the non-overlapping symbol $(|=>)$. This approach directs the LLMs' focus solely on identifying delays, simplifying their task. Moreover, VERT includes delayed assertion checks, facilitating the accurate extraction of clock cycle information from the source code.

**Miscapturing *if* Condition for *else* Branches:** VERT addresses the common omission of conditions in the *else/else-if* branches of *if-else* statements by exposing the model to diverse conditional structures, ensuring it accurately captures prior conditions when generating assertions. By incor-

porating examples where each *else* or *else-if* branch accounts for all preceding *if* conditions, the dataset trains LLMs to recognize the logical flow between branches. This enhances the model's ability to maintain logical consistency, leading to more accurate and complete assertion generation for conditional logic.

**Nested *If-Else* Conditions:**    To address the challenge of LLMs struggling with deeply nested *if-else* statements, we expanded our dataset to include complex, multi-level conditional structures. These examples focused specifically on scenarios where decision logic is nested beyond two levels, which is common in hardware designs but difficult for LLMs to handle. By providing a diverse set of deeply nested *if-else* conditions, we aim to enhance the LLM's ability to better recognize how each layer of decision-making is dependent on the preceding conditions. This approach ensures that the LLM generates assertions for each nested block without oversimplifying the logic or missing critical conditions in the inner branches. Furthermore, we refined the dataset to ensure that the LLM learns to correctly generate assertions even when the code lacks an explicit *else* branch, preserving the intended behavior of the original code. This ensures that if the condition is false, no action is required, and the state remains unchanged. However, LLMs can struggle with this distinction, often generating incorrect assertions by either assuming an implicit *else* branch or failing to account for the absence of any action when the condition evaluates to false. This process helps the model handle nested structures more effectively, producing accurate and logically consistent assertions for even the most complex hardware designs.

**Handling Long Conditions:**    To address the challenge of generating accurate assertions for long and complex conditions, we expanded the dataset to include a variety of cases where multiple conditions and operators must be evaluated simultaneously. These conditions often involve a combination of AND, OR, and NOT operators across several variables, making it essential for the model to handle intricate logical relationships. By exposing the LLM to examples that require the correct ordering and evaluation of these operators, VERT helps it learn to generate assertions that accurately reflect the complexity of the source code. *This approach ensures that all logical paths are captured in the assertions, avoiding the common pitfall of oversimplifying or omitting important parts of the condition*. The result is a more precise handling of extended logic chains, leading to fewer errors in assertion generation for complex hardware designs.

## 4.2 DATASET COMPOSITION

VERT comprises 20,000 samples, categorized based on the structural elements of SystemVerilog code and the nature of the assertions generated. We carefully divide VERT among various categories to ensure comprehensive coverage of the conditions encountered in hardware verification while addressing the weaknesses of current SoTA LLMs in generating assertions.

**Data Source and Cleanup:**    We compile a comprehensive list of variable names for VERT by extracting variables from hardware modules in various open-source Hardware Description Language (HDL) projects. As shown in Figure 1a, these variables are sourced from a diverse set of projects, including *BOOM-core* Zhao et al. (2020), *rocket-chip* Lee et al. (2016), and *XiangShan* Xu et al. (2022), each contributing over 150 variables to the dataset. *BOOM-core* leads with approximately 500 variables, while *rocket-chip* and *XiangShan* contribute around 450 variables each. By drawing from a diverse range of open-source modules, we ensure the model is exposed to various real-world scenarios. Many System-on-Chip (SoC) designs frequently reuse IP blocks from the same vendors, resulting in overlapping variable names. Similar IP blocks, such as various implementations of AES encryption, often perform identical operations, further contributing to naming redundancies. This reuse of IP, prevalent in both open-source and commercial SoCs, creates a degree of homogeneity in the design landscape, making it challenging to differentiate between components. To mitigate this issue and prevent overfitting to specific naming conventions or operations, we introduce randomly generated variables into the dataset, ensuring greater diversity and robustness in handling various designs. Once the variable list is compiled, it is cleaned up by removing duplicates, resolving inconsistencies, and verifying syntactic correctness. This ensures the model is exposed to various real-world hardware design scenarios while avoiding overfitting.

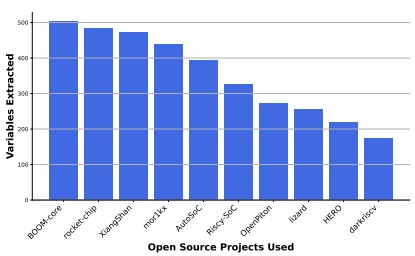

(a) Distribution of variables extracted from open-source HDL projects.

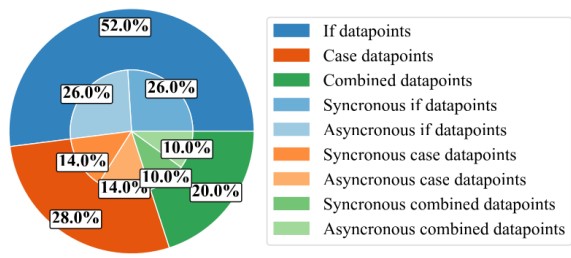

(b) Composition of our dataset.

Figure 1: Dataset source distribution and composition.

**Rationale Behind Data Composition:** As illustrated in Figure 1b, the largest portion of the dataset, comprising 52%, consists of *if-else* statements. This focus stems from the challenges LLMs like GPT-4o often face in generating accurate assertions for nested *if* structures, as discussed in Section 3. Building upon the intuition presented in Section 4.1, we structured the dataset to prioritize complex conditional scenarios. The complexity and layering of conditions in nested *if* statements frequently lead to errors, making them more problematic than other conditional structures. To address these issues, we emphasize *if-else* statements in our dataset. In contrast, case statements make up 28% of the dataset. Although commonly used to represent signals in hardware design, we encounter fewer difficulties when generating assertions for case-based logic, which accounts for their smaller proportion. Furthermore, 20% of the dataset includes combined statements, where *if* and *case* statements are intertwined to form more complex conditions. These mixed scenarios are included due to the added complexity, which presents challenges for LLMs when generating accurate assertions. We also include an even distribution of asynchronous and synchronous assertions in our dataset. This is crucial because for LLMs to capture clock cycle delays accurately, they must correctly interpret which signals in *if-else* or *case* statements are clock-sensitive. By providing a mix of both types of assertions, we ensure that the models learn to differentiate between immediate and clocked responses, enabling more accurate assertion generation in clock-sensitive hardware designs.

**Completeness:** To build upon the analysis from Figure 1b, where we emphasized the inclusion of various conditional structures in the dataset, it is important to highlight how these structures are integral to formulating assertions. When an assertion is formulated, the conditional structure of the function is required. These structures are constructed using *if-else* blocks, *case*, and *ternary* operators. The sensitivity (*i.e.,* when to check values for assertions) is taken from the *always* block. For instance, `always @(posedge clk_i)` denotes that the values should be checked at the rising edge of the `clk_i` signal. Our dataset contains all types of *always* blocks used in hardware design codes (*i.e.,* `always`, `always_ff`, `always_comb`). Other code components like *for* loops do not contribute to the formulation of assertions.

### 4.3 SYNTHETIC GENERATION OF ASSERTIONS

The dataset was synthetically generated to address the variability in how different repositories and projects formulate assertions. Many open-source repositories employ custom or project-specific assertion structures, leading to inconsistencies across sources. This lack of standardization makes it challenging to compile a cohesive dataset using only real-world examples. Moreover, relying solely on real-world data would not provide a sufficient number of consistent assertion structures for an LLM to effectively learn how to generate assertions from source code. Therefore, synthetic data is essential to create a comprehensive and uniform dataset suitable for training.

Generating the synthetic data involves creating a comprehensive set of conditions based on the cleaned variable list. These conditions serve as the foundation for creating structured code blocks, along with their corresponding assertions. By dynamically generating these conditions, we can ensure that the model is exposed to a wide array of patterns, preventing it from overly relying on specific naming conventions or design features commonly encountered in SoC components.

Figure 2 showcases the generation of synthetic *case* statements and their corresponding assertions. The process operates by extracting select lines from a dataset of variables and conditions, and for each line, it constructs a Verilog-style *case* block. It selects unique conditions and populates assignment operations. Since the conditional statements and assignment operation are known during dataset generation, the assertions can be constructed based on these conditions, ensuring consistency. In this process, the assertions are triggered on the rising edge of the clock (as indicated by the `@posedge clk_i` in the source code), ensuring that the logic is evaluated synchronously. The selected *case* checks the assigned condition, while subsequent cases ensure the appropriate actions for other input values. The default clause handles situations where none of the specified cases are met. Each *case* condition is followed by a delay to account for signal propagation and verify that the expected logic occurs at the correct time. Assertions for unselected cases confirm that invalid branches are not mistakenly triggered, ensuring the default behavior is correctly executed when applicable. Further examples of synthetic assertions are provided in appendix A.1.

Figure 2: Generation of assertions from Case Statements.

Figure 3 demonstrates how a hierarchy of asynchronous *if-else* conditions are systematically transformed into assertions that verify the correctness of combinational logic. Since there is a combinatorial block (`always_comb`) in the source code, the assertion created is asynchronous, hence devoid of a clock signal. The initial condition checks the first case, while nested conditions introduce additional layers of complexity. The *else-if* and *else* clauses account for alternative scenarios when the previous conditions are unsatisfied. In this logical flow, nested conditions are connected using an `AND` relation, requiring all specified conditions to be true for their corresponding assertions to activate. For the *else-if* and *else* branches, previous conditions are negated, ensuring the new condition only holds when prior conditions are false. This comprehensive approach effectively tests both *if* and *else* branches within the *if-else* block, providing thorough coverage of all possible logical states.

Figure 3: Generation of assertions from If statements.

## 5 RESULTS

### 5.1 EXPERIMENTAL SETUP

We evaluate VERT through a two-phase methodology. In the first phase, we fine-tune **Llama 3.1 8B** and **DeepSeek Coder 6.7B** and assess the syntactic and functional correctness of the assertions generated by these models. In the second phase, we evaluate whether the generated assertions accurately

describe the design's functionality and the coverage they achieve. The accuracy of representing the functionalities is tested using mutation testing, where multiple design variants with altered functionalities (mutants) are created to assess whether the assertions accurately capture design behavior by triggering appropriately in these mutants Iman et al. (2024). For coverage analysis, we employ Complete Path Coverage (CPC) as the metric, ensuring all independent paths in the design automaton are traversed Tong et al. (2010). Formal verification tools such as Cadence JasperGold and simulation tools like Xilinx Vivado are utilized throughout both phases to validate correctness, representational accuracy, and achieving up to 100% coverage.

Furthermore, since we cannot fine-tune GPT-4o due to it being a proprietary model, we compare the open-source fine-tuned models to GPT-4o to highlight the effectiveness of VERT. To assess their ability to generate code across diverse coding conventions and design principles, we test them on three open-source SoC designs—OpenTitan ope (2024), CVA6 Zaruba & Benini (2019), and Pulpissimo Schiavone et al. (2018).

The evaluation focuses on three primary metrics: the total number of generated assertions, the percentage of syntactically correct assertions, and the percentage of functionally correct assertions. Syntactic correctness refers to adherence to hardware description language standards, while functional correctness indicates that the assertions accurately reflect the intended hardware behavior (further elaborated in Appendix A.4). Through this comprehensive evaluation, we ensure that the generated assertions not only meet syntactical and functional criteria but also enhance the quality of verification for hardware IPs.

## 5.2 EVALUATION RESULTS

Table 1: Performance Comparison of base and fine-tuned models on assertion generation across various hardware IP benchmarks.

| Models | Benchmark/ Hardware IP | Generated Assertions | | Syntactically Correct Assertions (%) | | Functionally Correct Assertions (%) | |
|---|---|---|---|---|---|---|---|
| | | Base Model | Fine-Tuned Model | Base Model | Fine-Tuned Model | Base Model | Fine-Tuned Model |
| Llama 3.1 | OpenTitan/AES | 212 | 125 | 35.84 | 88.70 | 8.02 | 83.48 |
| | OpenTitan/I2C | 149 | 126 | 29.53 | 83.33 | 9.39 | 83.33 |
| | OpenTitan/LC CTRL | 26 | 19 | 23.07 | 89.47 | 7.69 | 89.47 |
| | OpenTitan/ADC CTRL | 63 | 32 | 17.46 | 100.00 | 9.52 | 100.00 |
| | CVA6/Frontend | 17 | 13 | 41.18 | 92.31 | 11.76 | 92.31 |
| | CVA6/Decode&Issue | 31 | 34 | 22.58 | 100.00 | 6.45 | 100.00 |
| | CVA6/Execute | 110 | 105 | 25.55 | 91.43 | 5.45 | 91.43 |
| | CVA6/Commit | 70 | 79 | 38.57 | 89.87 | 10 | 89.87 |
| | CVA6/Controller&Top | 73 | 68 | 34.24 | 95.59 | 5.48 | 95.59 |
| | Pulpissimo/APB | 15 | 19 | 53.33 | 89.47 | 53.33 | 89.47 |
| | Pulpissimo/RISCV | 19 | 15 | 21.05 | 93.33 | 21.05 | 93.33 |
| | Pulpissimo/debug_unit | 6 | 11 | 16.67 | 100.00 | 16.67 | 100.00 |
| DeepSeek Coder | OpenTitan/AES | 148 | 157 | 10.81 | 94.90 | 6.08 | 93.63 |
| | OpenTitan/I2C | 132 | 124 | 12.12 | 97.58 | 8.33 | 97.58 |
| | OpenTitan/LC CTRL | 21 | 19 | 14.25 | 100.00 | 9.52 | 100.00 |
| | OpenTitan/ADC CTRL | 32 | 32 | 6.25 | 100.00 | 0 | 96.88 |
| | CVA6/Frontend | 16 | 14 | 56.25 | 92.86 | 37.5 | 92.86 |
| | CVA6/Decode&Issue | 37 | 32 | 18.92 | 100.00 | 13.51 | 100.00 |
| | CVA6/Execute | 91 | 99 | 26.37 | 97.98 | 20.88 | 97.98 |
| | CVA6/Commit | 97 | 93 | 21.65 | 89.25 | 17.53 | 89.25 |
| | CVA6/Controller&Top | 82 | 76 | 21.95 | 89.47 | 15.85 | 89.47 |
| | Pulpissimo/APB | 25 | 19 | 24.00 | 100.00 | 24.00 | 100.00 |
| | Pulpissimo/RISCV | 13 | 15 | 23.08 | 100.00 | 23.08 | 100.00 |
| | Pulpissimo/debug_unit | 11 | 11 | 15.38 | 100.00 | 15.38 | 100.00 |

Table 1 compares the performance of the base and fine-tuned versions of the Llama 3.1 and DeepSeek Coder 6.7B models across various hardware IP benchmarks. The first column of the table lists the benchmark name, such as OpenTitan/AES, where the SoC name (OpenTitan) is followed by the specific IP name (AES). The subsequent columns display the number of assertions generated and the percentage of those assertions that are both syntactically and functionally correct for both base and fine-tuned models. The table is organized by model type, with performance metrics broken

down for each hardware IP block. Examples that highlight the improvements in the accuracy of assertion generation after fine-tuning the models are provided in appendix A.3.

Both the Llama 3.1 and DeepSeek Coder models demonstrated significant improvements over the base models following fine-tuning, with some benchmarks showing drastic gains. For Llama 3.1, syntactic correctness saw a maximum improvement of up to 83.33%. Similarly, the functional correctness showed a maximum increase of 93.55%.

The DeepSeek Coder model exhibited similarly substantial improvements. For instance, syntactic correctness improved as much as 93.75% (from 6.25% to 100%), and functional correctness increased up to 96.88% (from 0% to 96.88%). These results highlight the effectiveness of fine-tuning in improving the models' ability to generate accurate hardware assertions.

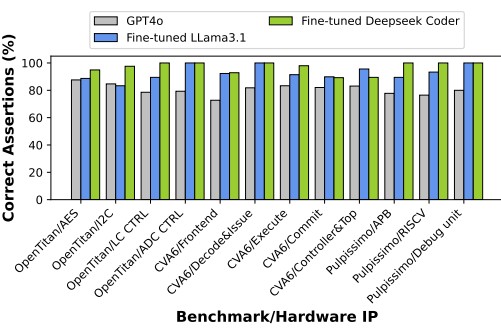 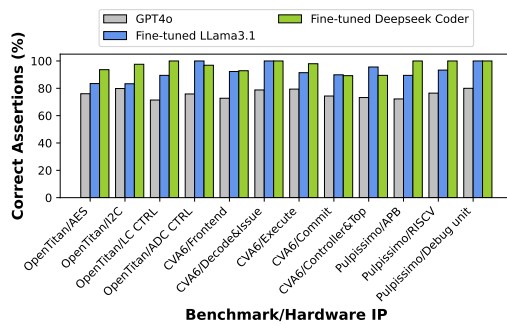

(a) Comparison of GPT-4o and Fine-tuned model in Syntactically correct assertions.

(b) Comparison of GPT-4o and Fine-tuned model in Functionally correct assertions.

Figure 4: Comparison of GPT-4o and Fine-Tuned Model Performance.

To illustrate VERT's effectiveness, we compare fine-tuned versions of the DeepSeek Coder 6.7B and Llama 3.1 8B model with GPT-4o. Figure 4a and Figure 4b show the syntactic and functional correctness of assertions generated by GPT-4o and the Fine-Tuned Llama 3.1 and Deepseek Coder models across various hardware benchmarks. The X-axis represents the benchmark SoC with its corresponding IP, such as OpenTitan AES, OpenTitan I2C, OpenTitan LC CTRL, and CVA6/Frontend, where assertions are evaluated. The Y-axis displays the percentage of correct assertions, indicating how reliably each model generated the assertions for each benchmark.

Figure 4a shows that Fine-tuned Llama 3.1 and Fine-tuned Deepseek Coder models significantly outperform GPT-4o by up to 20.69% in generating syntactically correct assertions. In Figure 4b, both Llama 3.1 and Deepseek Coder again outperformed GPT-4o by as much as 24.14% and 21.02% respectively, with functionally correct assertion in modules such as CVA6/Decode&Issue and Pulpissimo/Debug unit. These results emphasize that LLMs fine-tuned on VERT enhance not only syntactic correctness but also the functional reliability of the generated hardware assertions.

## 6 CONCLUSION

In this paper, we introduce VERT, a novel open-source dataset tailored to automate the generation of SystemVerilog assertions, enabling a more scalable and efficient hardware verification process using LLMs. By systematically fine-tuning popular models such as DeepSeek Coder and LLaMA 3.1 on our dataset, we achieved substantial improvements in both syntactical accuracy and functional correctness of generated assertions across real-world SoCs, including OpenTitan, CVA6, and Pulpissimo. Our evaluation demonstrated the adaptability of these LLMs, fine-tuned with VERT, furnishing up to a 96.88% improvement in both functional and syntactical correctness over base models and up to 24.14% over GPT-4o. This work is the first to demonstrate the potential of combining domain-specific datasets with advanced LLMs to address the enhanced challenges of modern hardware verification. In the future, we will focus on expanding the dataset to cover more intricate design patterns and hardware architectures, as well as improving model performance in handling asynchronous and synchronous conditions. Moreover, we aim to integrate our approach with industry-standard functional verification tools to streamline the hardware verification process.

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

CONTENTS

# A    APPENDIX

## A.1   DATASET EXAMPLES

Figure 5 presents synthetic SystemVerilog code snippets and their formal assertions from our dataset. The left column shows representative code blocks with conditional and case-based logic. The top-left snippet is from the "if-else" dataset, where the assertion checks that when $!cfg\_7$ is true and $hw\_13$ && $reg\_14$ || $core\_17$ holds, $chip\_9$ must equal $fsm\_18$. This ensures correct signal assignment following the logic during simulation or formal verification.

The middle-left snippet, from the "case" dataset, assigns $chip\_12$ to $tx\_10$ if $flag\_register\_18$ equals $5'b10011$. The corresponding assertion ensures that this assignment occurs correctly. The bottom-left snippet showcases a case statement where if $flag\_register\_18$ is $7'h26$ and the compound condition ($rx\_6$ || $rx\_1$ || $hw\_7$) holds, $cfg\_20$ and $reg\_3$ are assigned the expected values, formalized by an assertion.

```
1 ...
2 if ( !cfg_7 ) begin
3 ...
4  else if ( hw_13  && reg_14  || core_17 )
          begin
5    chip_9 = fsm_18;
6 ...
```

```
1 ...
2 property name;
3  (!cfg_7) && (hw_13 && reg_14 || core_17)
4  |-> chip_9 == fsm_18;
5 endproperty
6 ...
```

```
1 ...
2 case ( flag_register_18 )
3    5'b10011 : begin
4       chip_12 = tx_10;
5    end
6 ...
```

```
1 ...
2 property name;
3 ( flag_register_18 ) == ( 5'b10011 )
4 |-> chip_12 == tx_10 ;
5 endproperty
6 ...
```

```
1 ...
2 case ( flag_register_18 )
3    7'h26 : begin
4 ...
5    if ( rx_6  || rx_1  || hw_7 ) begin
6       cfg_20 = err_20;
7       reg_3 = rst_15;
8 ...
```

```
1 ...
2 property name;
3 ( flag_register_18 ) == ( 7'h26 ) &&
4 ( rx_6  || rx_1  || hw_7 ) |->
5 cfg_20 == err_20 &&
6 reg_3 == rst_15 ;
7 endproperty
8 ...
```

Figure 5: Examples of code blocks (left) and their corresponding assertions (right) in our dataset.

## A.2   FINE-TUNING

We utilized the hyperparameters detailed in Table 2, leveraging the Low-Rank Adapter (LoRA) technique with a rank and alpha of 256 to optimize the model for hardware verification tasks with minimal computational overhead. This configuration allowed for efficient low-rank updates, preserving the core model's behavior while integrating task-specific adjustments. A max-

imum sequence length of 4096 tokens was employed to accommodate longer logic and condition sequences in assertion generation, with training constrained to 3 epochs to avoid overfitting.

We selected a batch size of 64 and a learning rate of $1.00e^{-04}$ for computational efficiency and stable convergence, with Bf16 precision enhancing training speed without compromising accuracy. By targeting the `Query`, `Key`, `Value`, `Output`, and `Gate` layers, we effectively adapted the model while updating only 4-6% of its parameters, thus optimizing performance while controlling computational costs.

| Hyperparameter | Value |
|---|---|
| Lora RANK | 256 |
| Lora alpha | 256 |
| Maximum sequence length | 4096 |
| Epochs | 3 |
| Batch | 64 |
| Learning_rate | $1.00e^{-04}$ |
| Training Presicion | Bf16 |

Table 2: Training Hyperparameters

## A.3 EXAMPLES
COMPARING FINE-TUNED MODEL TO BASE MODEL

### A.3.1 OPENTITAN ASSERTION EXAMPLES

---

DeepSeek Coder input prompt

### Instruction: Generate a list of synchronous SystemVerilog assertions executing at posedge clk_i from the following code:

```
1    if (!rst_ni) begin
2      edn_req_hold_q <= '0;
3    end else begin
4      edn_req_hold_q <= edn_req_hold_d;
5    end
```

### Response:

---

DeepSeek Coder base model response

The given code is a synchronous systemverilog code for a flip-flop. It is a simple register that holds its value on the rising edge of the clock signal 'clk_i'. The reset signal 'rst_ni' is used to clear the register.
The synchronous systemverilog assertion executing at posedge clk_i would be:

```
1 assert property (
2   @(posedge clk_i) disable iff (~rst_ni)
3   (edn_req_hold_q == $past(edn_req_hold_d, 1))
4 );
```

This assertion checks that the value of 'edn_req_hold_q' is equal to the value of 'edn_req_hold_d' at the previous clock cycle. The '$past' function is used to access the value of 'edn_req_hold_d' from the previous cycle. The 'disable iff ( rst_ni)' part ensures that the assertion is not checked when 'rst_ni' is low.

---

In the base-model generated assertions, the chain of events depends on the reset signal, however, the assertion disables the 'rst_ni' signal. Moreover, the generated checking statement does not reflect the correct chain of events. 'edn_req_hold_q' should be assigned the current value of 'edn_req_hold_d', not the value it had in the previous clock cycle.

---

DeepSeek Coder fine-tuned model response

Here is the requested property

```
1 property ResetSynceotid; @(posedge clk_i) (!rst_ni)|-> edn_req_hold_q == '0; endproperty
2 property SyncReqeotid; @(posedge clk_i) ( rst_ni)|->edn_req_hold_q==edn_req_hold_d; endproperty
```

---

When generated by the same model fine-tuned on VERT, it generates the assertions with 'rst_ni' under consideration, and it gets the correct clock cycle.

---

**Llama 3.1 input prompt**

$< |eot\_id| >< |start\_header\_id| > user < |end\_header\_id| >$ Generate a list of synchronous systemverilog assertion executing at posedge clk_i from the following code:

```
1    if (!rst_ni) begin
2      lc_state_valid_q <= 1'b0;
3    end else begin
4      lc_state_valid_q <= lc_state_valid_d;
5    end
```

$< |eot\_id| >< |start\_header\_id| > assistant < |end\_header\_id| >$

---

**Llama 3.1 base model response**

Here is the SystemVerilog property that corresponds to the given code:

```
1 property lc_state_valid_q_posedge;
2 @(posedge clk_i) ($rose(!rst_ni)|->lc_state_valid_q===1'b0)|->lc_state_valid_q===
        lc_state_valid_d;
3 endproperty
```

This property states that when the clock edge occurs, if the reset signal is asserted (i.e., '!rst_ni' is true), then 'lc_state_valid_q' should be 0. If the reset signal is not asserted, then 'lc_state_valid_q' should be equal to 'lc_state_valid_d'.

Note that the 'rose' function is used to check if the reset signal is asserted (i.e., goes from 0 to 1) at the current clock edge.

---

Here, the base-model generated assertion does not get the chain of events right. The if-else block describes that if there is a reset invoked, 'lc_state_valid_q' is assigned 0. Otherwise, it should be assigned the current value of 'lc_state_valid_d.' However, the generated assertion asserts that if '!rst_ni' is risen, 'lc_state_valid_q' should be assigned 0, and in the next clock cycle, it should be assigned 'lc_state_valid_d.'

---

**Llama 3.1 fine-tuned model response**

Here is the requested property

```
1 property ResetSync1;@(posedge clk_i) (!rst_ni)|->lc_state_valid_q==1'b0; endproperty
2 property ResetSync2;@(posedge clk_i) (rst_ni)|->lc_state_valid_q==lc_state_valid_d; endproperty
```

### A.3.2 CVA6 ASSERTION EXAMPLES

---

**DeepSeek Coder input prompt**

### Instruction:Generate a list of asynchronous systemverilog assertion from the following code

```
1    if (CVA6Cfg.RVH) begin
2      if (CVA6Cfg.IS_XLEN64) begin : gen_vsstatus_64read
3        assign vsstatus_extended = vsstatus_q[CVA6Cfg.XLEN-1:0];
4      end else begin : gen_vsstatus_32read
5        assign vsstatus_extended = {vsstatus_q.sd, vsstatus_q.wpri3[7:0], vsstatus_q[22:0]};
6      end
7    end else begin
8      assign vsstatus_extended = '0;
9    end
```

### Response:

---

DeepSeek Coder base model response

The code you provided is a SystemVerilog assertion. However, it does not contain any asynchronous assertions. Asynchronous assertions are used to check the conditions at different times. In your code, there are no conditions that would be checked asynchronously.

If you want to create an asynchronous assertion, you would need to use the '`$rose`' or '`$fell`' functions to check for changes in the signal. However, this is not applicable in your code as there are no changes in the signal that you are checking.

Here is an example of an asynchronous assertion:

```
1  always @(posedge clk or posedge reset) begin
2    if (reset) begin
3      // reset condition
4    end
5    else begin
6      // check condition
7      if ($rose(condition)) begin
8        // do something
9      end
10   end
11 end
```

In this example, '`$rose(condition)`' checks if 'condition' rose (changed from 0 to 1) at the next clock edge.

In this case, the base model generated an always block that does not assert anything.

DeepSeek Coder fine-tuned model response

Here is the requested property

```
1
2   property ValidSynceotid; (CVA6Cfg.RVH) |-> (CVA6Cfg.IS_XLEN64) |-> vsstatus_extended ==
        vsstatus_q[CVA6Cfg.XLEN-1:0] ;endproperty
3
4   property ValidSynceotid; (CVA6Cfg.RVH) |->  (CVA6Cfg.IS_XLEN64) |-> vsstatus_extended == {
        vsstatus_q.sd, vsstatus_q.wpri3[7:0], vsstatus_q[22:0]} ; endproperty
5
6   property SafeReseteotid; (CVA6Cfg.RVH) |-> vsstatus_extended == '0; endproperty
```

Llama 3.1 input prompt

$<|start\_header\_id|> user <|end\_header\_id|>$ Generate a list of asynchronous systemverilog assertion from the following code

```
1      if (CVA6Cfg.RVS) begin
2        satp = satp_q;
3      end
```

$<|eot\_id|><|start\_header\_id|> assistant <|end\_header\_id|>$

Llama 3.1 base model response

**Property: satp is updated when CVA6Cfg.RVS is asserted**
systemverilog

```
1  property p_satp_updated;
2    @(posedge clk) ($rose(CVA6Cfg.RVS) |=> satp === satp_q);
3  endproperty
```

In this case, while the base model generated a syntactically correct property, it mistook the check condition as an edge-triggered event instead of a level-triggered event. Furthermore, it also got the clock cycle wrong. Instead of defining the property for the current clock cycle, it used the next clock cycle.

---

**Llama 3.1 fine-tuned model response**

```
1  property SyncRegeotid; (CVA6Cfg.RVS) |-> satp == satp_q ;endproperty
```

---

### A.3.3 PULPISSIMO ASSERTION EXAMPLES

---

**DeepSeek Coder input prompt**

### Instruction:Generate a list of asynchronous systemverilog assertion from the following code

```
1      case (rdata_sel_q)
2        RD_CSR:  debug_rdata_o = csr_rdata_i;
3        RD_GPR:  debug_rdata_o = regfile_rdata_i;
4        RD_DBGA: debug_rdata_o = dbg_rdata;
5        RD_DBGS: debug_rdata_o = dbg_rdata;
6      endcase
```

### Response:

---

**DeepSeek Coder base model response**

The code you provided is a SystemVerilog case statement. It seems like you want to generate a list of asynchronous assertions for this code. However, SystemVerilog does not support assertions in the same way as other languages (like C, Java, etc.). Assertions in SystemVerilog are used to check the design during compile time or run time.

If you want to generate asynchronous assertions, you would typically use the `$assertion_off,` `$assertion_on,` `$assume,` `$cover,` or `$fell,` `$rose` constructs.

However, without more context, it's hard to provide a more specific answer. If you could provide more information about the expected behavior or the conditions you want to check, I could give a more precise answer.

For example, if you want to ensure that 'rdata_sel_q' is always valid (i.e., it's either 'RD_CSR,' 'RD_GPR,' 'RD_DBGA,' or 'RD_DBGS'), you could use an assertion like this:

```
1  always @(*) begin
2    $assert_on;
3    assert (rdata_sel_q inside {RD_CSR, RD_GPR, RD_DBGA, RD_DBGS}) else $error("Invalid
        rdata_sel_q value");
4  end
```

---

Here, the base model only checks whether the value of 'rdata_sel_q' is within the defined parameters but not the operations.

---

**DeepSeek Coder fine-tuned model response**

Here is the requested property

```
1  property DataSynceotid; (rdata_sel_q)==(RD_DBGS)|->debug_rdata_o==dbg_rdata; endproperty
2  property ValidRdeotid; (rdata_sel_q)==(RD_CSR)|->debug_rdata_o==csr_rdata_i; endproperty
3  property ValidRdeotid; (rdata_sel_q)==(RD_GPR)|->debug_rdata_o==regfile_rdata_i; endproperty
4  property ValidRdeotid; (rdata_sel_q)==(RD_DBGA)|->debug_rdata_o==dbg_rdata; endproperty
```

---

**Llama 3.1 input prompt**

$< |eot\_id| >< |start\_header\_id| > user < |end\_header\_id| >$ Generate a list of synchronous systemverilog assertion executing at posedge HCLK from the following code:

```
1        if(~HRESETn)
2        begin
3            r_status  <=  'h0;
4        end
5        else
6        begin
7            if (s_rise_int )
8            begin
9                r_status  <= r_status | s_is_int_all;
10           end
11           else if (PSEL && PENABLE &&!PWRITE && (s_apb_addr == 'REG_INTSTATUS))
12           begin
13               r_status  <=  'h0;
14           end
15       end
16   end
```

$< |eot\_id| >< |start\_header\_id| > assistant < |end\_header\_id| >$

**Llama 3.1 base model response**

Here is the list of synchronous SystemVerilog assertions that can be generated from the given code, executing at posedge HCLK:

```
1 property p_reset_default_value;
2     @(posedge HCLK) disable iff (~HRESETn) $rose(r_status) == 1;
3 endproperty
```

```
1 property p_int_status_set;
2     @(posedge HCLK) disable iff (~HRESETn) (s_rise_int) |=> (r_status | s_is_int_all);
3 endproperty
```

```
1 property p_apb_reset_int_status;
2     @(posedge HCLK) disable iff (~HRESETn) (PSEL && PENABLE &&!PWRITE && (s_apb_addr ==
      'REG_INTSTATUS)) |=> (r_status == 'h0);
3 endproperty
```

Note: The 'disable iff' keyword is used to disable the property when the condition is false. The 'rose' function is used to check if the value of 'r_status' has risen (i.e., changed from 0 to 1) at the current clock edge. The '—=¿' operator is used to specify the implication relationship between the antecedent and the consequent.

Here, the assertions miss the correct clock cycle for the assertions. Furthermore, it missed the nested conditional blocks under 'HRESETn,' which should have put the nested condition in an "AND" relation with 'HRESETn.'

**Llama 3.1 fine-tuned model response**

Here is the requested property

```
1 property ResetSync1; @(posedge clk_i) (!rst_ni)|->lc_state_valid_q==1'b0; endproperty
2 property ResetSync2; @(posedge clk_i) (rst_ni)|->lc_state_valid_q==lc_state_valid_d; endproperty
```

## A.4 DIFFERENCE BETWEEN FUNCTIONAL AND SYNTACTICAL ASSERTIONS

---

Syntactical and Functional Errors in Assertions.

Design Source Code:

```systemverilog
always_ff @(posedge clk_aon_i or negedge rst_aon_ni) begin
    if (!rst_aon_ni) begin
        wakeup_timer_cnt_q    <= '0;
    end
    else if (wakeup_timer_cnt_clr || cfg_fsm_rst_i || trigger_h2l) begin
        wakeup_timer_cnt_q <= '0;
    end else begin
        wakeup_timer_cnt_q <= wakeup_timer_cnt_d;
    end
end
```

Here is the requested property

```systemverilog
property ResetTimer1; @(posedge clk_aon_i) (!rst_aon_ni)|->wakeup_timer_cnt_q=1'b0; endproperty
// A syntactically incorrect assertion
// Here, instead of using '==' symbol, '=' was used
property ResetTimer2;
@(posedge clk_aon_i)(wakeup_timer_cnt_clr||cfg_fsm_rst_i||trigger_h2l)|->wakeup_timer_cnt_q=='0;
endproperty
// A Functionally incorrect assertion
// Here, the generated assertion is missed capturing the 'if' condition.
```

---

Since formal tools like Cadence JasperGold assume that asynchronous reset signals remain inactive during execution Miftah et al. (2024), we use simulation in Xilinx Vivado to verify assertions involving asynchronous resets and ensure the asserted properties can be covered. Next, we validate the functional correctness of the assertions by performing formal and simulation verification runs, validating that the assertions can be triggered and observed during testing. This two-stage process ensures that only syntactically correct and functionally valid assertions are retained in the design. Textbox A.4 illustrates the difference between syntactical and functional errors in assertions within a hardware design context. It presents two issues that can arise when writing assertions to verify system behavior.

**Design Source Code:** The provided SystemVerilog code shows an always_ff block, triggered by either the rising edge of clk_aon_i or the falling edge of rst_aon_ni. The block resets or updates the value of wakeup_timer_cnt_q based on certain conditions:

1. If rst_aon_ni is low (reset active), the counter is set to zero.

2. If wakeup_timer_cnt_clr or certain other signals are asserted, the counter is reset.

3. Otherwise, the counter is updated with a new value from wakeup_timer_cnt_d.

**Assertions:** Two properties are presented, each demonstrating a different type of error:

- **Syntactical Error:**
  In property ResetTimer1, the assertion attempts to check if the counter is reset when rst_aon_ni is low. However, it contains a **syntactical error**: instead of using the comparison operator == to check if wakeup_timer_cnt_q equals zero, the assignment operator = is mistakenly used. This would result in a syntax error during compilation.

- **Functional Error:**
  In property ResetTimer2, while the syntax is correct, the assertion misses an essential condition. It checks whether the counter is reset when the clear signal or related signals are asserted. However, it fails to include the reset condition (i.e., if (!rst_aon_ni)), resulting in a **functional error** because the assertion does not fully capture the intended behavior of the design, specifically missing the if condition from the original source code.

**Key Point:** We highlight how both types of errors (syntactical and functional) can undermine the correctness of assertions, either by preventing successful compilation (in the case of syntax errors) or by failing to accurately verify the design's intended behavior (in the case of functional errors).

## A.5 Examples of LLMs Fine-tuned on VERT Addressing GPT-4o Issues and Comparison With Human Experts

In this section, we show how the intuition in dataset formulation, as discussed in Section 4.1 of our paper, translates into the effectiveness of the assertions generated by LLMs fine-tuned on VERT. Furthermore, we also provide a comparison of these models with human experts. These scenarios include (1) clock cycle and pre-condition capturing, (2) relation representation for nested conditional blocks, and (3) handling long conditions. Each section begins with the relevant code snippet, followed by the assertions generated by each approach.

---

A.1: Clock Cycle and Pre-condition Capturing in GPT-4o VS Vs LLMs fine tuned on VERT.

Generate a list of synchronous SystemVerilog assertion executing at (*posedge* clk_i) from the following code:

```
10  always @(posedge clk_i or negedge rst_ni) begin
11      if (~rst_ni) begin
12          wr_state_q <= IDLE;
13          wr_cnt_q   <= '0;
14      end else begin
15          wr_state_q <= wr_state_d;
16          wr_cnt_q   <= wr_cnt_d;
17      end
18  end
```

GP4o Response:

```
1  ...
2  // Assertion to ensure that wr_state_q follows wr_state_d when rst_ni is high
3  assert property (@(posedge clk_i) (rst_ni && (wr_state_d !== IDLE)) |=> (wr_state_q ==
        wr_state_d))
4    else $error("State transition failure: wr_state_q did not follow wr_state_d.");
5
6  // Assertion to ensure that wr_cnt_q follows wr_cnt_d when rst_ni is high
7  assert property (@(posedge clk_i) (rst_ni && (wr_cnt_d !== '0)) |=> (wr_cnt_q == wr_cnt_d))
8    else $error("Counter transition failure: wr_cnt_q did not follow wr_cnt_d.");
9
10 // Assertion to check that the state and count do not update during reset
11 assert property (@(posedge clk_i) (~rst_ni) |=> (wr_state_q == IDLE && wr_cnt_q == '0))
12   else $error("Reset condition not maintained: wr_state_q or wr_cnt_q changed during reset.");
13 ...
```

VERT Llama 3 Response:

```
1
2  property ResetSynceotid; @(posedge clk_i) ( ~rst_ni ) |-> wr_state_q == IDLE && wr_cnt_q == '0 ;
        endproperty
3
4  property ResetSynceotid; @(posedge clk_i)  ( rst_ni )  |-> wr_state_q == wr_state_d && wr_cnt_q
        == wr_cnt_d; endproperty
```

VERT DeepSeek Response:

```
1  property ResetSynceotid; @(posedge clk_i) ( ~rst_ni ) |-> wr_state_q == IDLE && wr_cnt_q == '0 ;
        endproperty
2
3  property ResetSynceotid; @(posedge clk_i)  ( rst_ni )  |-> wr_state_q == wr_state_d && wr_cnt_q
        == wr_cnt_d; endproperty
```

---

### A.5.1 LLMs Fine-tuned on VERT vs. GPT-4o:

**Clock Cycle Misinterpretations:** We observe that LLMs often get confused between using the overlapping implication symbol ($|->$) and the non-overlapping symbol ($|=>$). To resolve clock cycle misinterpretation by LLMs, we standardized our format by using the overlapping implication symbol with a specified delay count and replacing the non-overlapping symbol. As shown in text box A.1 below, GPT-4o incorrectly uses the non-overlapping symbol, but both LLMs fine-tuned on VERT address this issue by correctly interpreting the clock cycle by using the overlapping implication symbol.

**Miscapturing if Condition for else Branches:** LLMs often overlook the previous if/else if conditions when generating assertions for the subsequent else or else if branches in conditional statements. VERT addresses the common omission of conditions in the else/else-if branches of if-else statements by exposing the model to diverse conditional structures. As shown in text box A.2 below, GPT-4o, in its second assertion, misses that the second assertion should verify that *monitor_enable* is *FALSE*

and $monitor\_enable\_q$ is *TRUE*. However, both LLMs fine-tuned on VERT correctly identify these conditions and include them in the fourth generated assertion.

---

A.2: Relation Representation for Nested Conditional Blocks in GPT-4o VS Vs LLMs fine tuned on VERT.

Generate a list of asynchronous SystemVerilog Assertion from the following code

```
1  if (!rst_ni) begin
2      bus_release_cnt <= '0;
3  end else if (monitor_enable &&!monitor_enable_q) begin
4      if (multi_controller_enable_i) begin
5          bus_release_cnt <= 30'(bus_inactive_timeout_i);
6      end
7  end else if (bus_release_cnt_load) begin
8      bus_release_cnt <= bus_release_cnt_sel;
9  end else if (bus_release_cnt_dec && (bus_release_cnt!= '0)) begin
10     bus_release_cnt <= bus_release_cnt - 1'b1;
11 end
```

GPT-4o Response:

```
1  ...
2  // Synchronous assertion for monitor enable signal transition
3  assert property (@(posedge clk) disable iff (!rst_ni)
4      (monitor_enable && !monitor_enable_q) |-> (multi_controller_enable_i ?
5      (bus_release_cnt == bus_inactive_timeout_i) : 1'b1)
6  ) else $fatal("Bus release count should be set to bus_inactive_timeout_i when monitor_enable
          rises and multi_controller_enable_i is high.");
7
8  // Synchronous assertion for bus_release_cnt_load condition
9  assert property (@(posedge clk) disable iff (!rst_ni)
10     (bus_release_cnt_load) |-> (bus_release_cnt == bus_release_cnt_sel)
11 ) else $fatal("Bus release count should be loaded with bus_release_cnt_sel when
          bus_release_cnt_load is asserted.");
12 ...
```

VERT Llama 3 Response:

```
1  property ResetSynceotid;  ( !rst_ni ) |-> bus_release_cnt == '0 ;endproperty
2
3  property ResetSynceotid; (  rst_ni ) && ( !monitor_enable || monitor_enable_q ) |->
       bus_release_cnt == '0 ;endproperty
4
5  property ResetSynceotid;  (  rst_ni ) &&  ( monitor_enable && !monitor_enable_q ) &&  (
       multi_controller_enable_i ) |-> bus_release_cnt == 30'(bus_inactive_timeout_i) ;endproperty
6
7  property ResetSynceotid;  (  rst_ni ) &&  ( !monitor_enable || monitor_enable_q ) && (
       bus_release_cnt_load ) |-> bus_release_cnt == bus_release_cnt_sel ;endproperty
8
9  property ResetSynceotid; (  rst_ni ) &&  ( !monitor_enable || monitor_enable_q ) && ( !
       bus_release_cnt_load ) && (  bus_release_cnt_dec && (bus_release_cnt != '0)) |->
       bus_release_cnt == (bus_release_cnt - 1'b1) ;endproperty
```

VERT DeepSeek Response:

```
1  property ResetSynceotid;  ( !rst_ni ) |-> bus_release_cnt == '0 ;endproperty
2
3  property ResetSynceotid;  (  rst_ni ) && ( !monitor_enable || monitor_enable_q ) |->
       bus_release_cnt == '0 ;endproperty
4
5  property ResetSynceotid;  (  rst_ni ) &&  ( monitor_enable && !monitor_enable_q ) &&  (
       multi_controller_enable_i ) |-> bus_release_cnt == 30'(bus_inactive_timeout_i) ;endproperty
6
7  property ResetSynceotid;  (  rst_ni ) &&  ( !monitor_enable || monitor_enable_q ) && (
       bus_release_cnt_load ) |-> bus_release_cnt == bus_release_cnt_sel ;endproperty
8
9  property ResetSynceotid;  (  rst_ni ) &&  ( !monitor_enable || monitor_enable_q ) && ( !
       bus_release_cnt_load ) && (  bus_release_cnt_dec && (bus_release_cnt != '0)) |->
       bus_release_cnt == (bus_release_cnt - 1'b1) ;endproperty
```

**Nested If-Else Conditions:** LLMs often struggle with generating assertions for deeply nested if-else conditions (i.e., blocks nested beyond two levels), which are common in hardware design. To address the challenge of LLMs struggling with deeply nested if-else statements, we expanded VERT to include complex, multi-level conditional structures. As illustrated by text box A.2 below, GPT-4o incorrectly represents the logic of a nested block. Specifically, it ignores the if condition and creates an assertion using a ternary operator, where the inner if condition is used as the test, the assignment is the *TRUE* branch, and *1'b1* is the *TRUE* branch. However, both LLMs fine-tuned on VERT to correctly identify the correct nested statement and their conditions.

---

A.3: Handling Long Conditions in GPT-4o Vs LLMs fine tuned on VERT.

Generate a list of asynchronous systemverilog assertion from the following code

```
1    if (state_sel_o      != mr_state_sel[i]      ||
2        add_rk_sel_o     != mr_add_rk_sel[i]     ||
3        key_full_sel_o   != mr_key_full_sel[i]   ||
4        key_dec_sel_o    != mr_key_dec_sel[i]    ||
5        key_words_sel_o  != mr_key_words_sel[i]  ||
6        round_key_sel_o  != mr_round_key_sel[i]) begin
7      mr_err = 1'b1;
8    end
```

GPT -4o Response:

```
1  ...
2  // Asynchronous assertion to check key_full_sel_o against mr_key_full_sel[i]
3  assert property ((key_full_sel_o != mr_key_full_sel[i])  |-> (mr_err == 1'b1))
4    else $error("Asynchronous check failure: key_full_sel_o does not match mr_key_full_sel[%0d]
         and mr_err is not asserted.", i);
5
6  // Asynchronous assertion to check key_dec_sel_o against mr_key_dec_sel[i]
7  assert property ((key_dec_sel_o != mr_key_dec_sel[i])  |-> (mr_err == 1'b1))
8    else $error("Asynchronous check failure: key_dec_sel_o does not match mr_key_dec_sel[%0d] and
         mr_err is not asserted.", i);
9
10 // Asynchronous assertion to check key_words_sel_o against mr_key_words_sel[i]
11 assert property ((key_words_sel_o != mr_key_words_sel[i]) |-> (mr_err == 1'b1))
12   else $error("Asynchronous check failure: key_words_sel_o does not match mr_key_words_sel[%0d]
         and mr_err is not asserted.", i);
13 ...
```

VERT Llama 3 Response:

```
1
2  property SyncErreotid; (state_sel_o     != mr_state_sel[i]      ||
3             add_rk_sel_o    != mr_add_rk_sel[i]     ||
4             key_full_sel_o  != mr_key_full_sel[i]   ||
5             key_dec_sel_o   != mr_key_dec_sel[i]    ||
6             key_words_sel_o != mr_key_words_sel[i]  ||
7             round_key_sel_o != mr_round_key_sel[i]) |-> mr_err = 1'b1 ;endproperty
```

VERT DeepSeek Response:

```
1   property SyncErreotid; (state_sel_o     != mr_state_sel[i]) ||
2      (add_rk_sel_o     != mr_add_rk_sel[i]) ||
3      (key_full_sel_o   != mr_key_full_sel[i]) ||
4      (key_dec_sel_o    != mr_key_dec_sel[i]) ||
5      (key_words_sel_o  != mr_key_words_sel[i]) ||
6      (round_key_sel_o  != mr_round_key_sel[i]) |-> mr_err == 1'b1 ;
7   endproperty
```

---

**Handling Long Conditions:** To address the challenge of generating accurate assertions for long and complex conditions, we expanded VERT to include a variety of cases where multiple conditions and operators must be evaluated simultaneously. As shown in text box A.3 below, the source code requires any one of six conditions to be met before raising the $mr\_err$ flag. However, the generated output by GPT- 4o creates three assertions for one flag change, each neglecting the other three conditions. This not only leads to incorrect assertions but also causes overhead in the number of assertions. However, both LLMs fine-tuned on VERT correctly generate a singular assertion that describes all the conditions and their relationships within the if-statement.

### A.5.2 LLMs FINE-TUNED ON VERT VS. HUMAN EXPERT:

Human experts and LLMs fine-tuned on VERT generate assertions differently. Experts often create simpler, more readable assertions, reflecting a preference for human-friendly formats. LLMs, on the other hand, produce more standardized logical expressions. This difference highlights style preferences rather than superiority. Notably, despite these stylistic differences, LLMs fine-tuned on VERT achieve assertion generation performance comparable to that of human experts. This is illustrated in the textbox A.4. For instance, in example 1, when verifying whether the `bus_release_cnt` signal has been reset, the LLM-generated assertion takes the form `bus_release_cnt == '0'`, while the human expert expresses the same operation as `!(|bus_release_cnt)`. The latter format, often favored by experts, is typically chosen for its ease of writing and brevity. Similarly, in example 2, since $mr\_err$ is a one-bit flag, the human expert chooses to verify whether the flag was raised by writing `mr_err` for the check, while the LLMs use the full expression, `mr_err == 1'b1`.

### A.4: Examples comparing LLMs Fine-tuned on VERT to Human Experts

**Example 1**

Source Code:

```
1  if (!rst_ni) begin
2              bus_release_cnt <= '0;
3  end else if (monitor_enable &&!monitor_enable_q) begin
4              if (multi_controller_enable_i) begin
5              bus_release_cnt <= 30'(bus_inactive_timeout_i);
6              end
7  end else if (bus_release_cnt_load) begin
8              bus_release_cnt <= bus_release_cnt_sel;
9  end else if (bus_release_cnt_dec && (bus_release_cnt!= '0)) begin
10             bus_release_cnt <= bus_release_cnt - 1'b1;
11 end
```

VERT Llama 3 Response:

```
1   property ResetSynceotid;
2   ( !rst_ni ) |-> bus_release_cnt == '0 ;
3   endproperty
4   property ResetSynceotid;
5   ( rst_ni ) && ( !monitor_enable || monitor_enable_q ) |-> bus_release_cnt == '0 ;
6   endproperty
7  property ResetSynceotid;
8   ( rst_ni ) &&  ( monitor_enable && !monitor_enable_q ) &&  (multi_controller_enable_i ) |->
       bus_release_cnt == 30'(bus_inactive_timeout_i);
9   endproperty
10 property ResetSynceotid;
11  ( rst_ni ) &&  ( !monitor_enable || monitor_enable_q ) && (  bus_release_cnt_load ) |->
       bus_release_cnt == bus_release_cnt_sel ;
12  endproperty
13 property ResetSynceotid;
14  ( rst_ni ) &&  ( !monitor_enable || monitor_enable_q ) && ( !bus_release_cnt_load ) &&  (
       bus_release_cnt_dec && (bus_release_cnt != '0)) |-> bus_release_cnt == (bus_release_cnt -
       1'b1) ;
15  endproperty
```

VERT DeepSeek Response:

```
1  property ResetSynceotid;
2  ( !rst_ni ) |-> bus_release_cnt == '0;
3   endproperty
4  property ResetSynceotid;
5  ( rst_ni ) && ( !monitor_enable || monitor_enable_q ) |-> bus_release_cnt == '0;
6   endproperty
7  property ResetSynceotid;
8  ( rst_ni ) &&  ( monitor_enable && !monitor_enable_q ) &&  ( multi_controller_enable_i ) |->
       bus_release_cnt == 30'(bus_inactive_timeout_i) ;
9   endproperty
10 property ResetSynceotid;
11  ( rst_ni ) &&  ( !monitor_enable || monitor_enable_q ) && (  bus_release_cnt_load ) |->
       bus_release_cnt == bus_release_cnt_sel ;
12  endproperty
13 property ResetSynceotid;
14  ( rst_ni ) &&  ( !monitor_enable || monitor_enable_q ) && ( !bus_release_cnt_load ) &&  (
       bus_release_cnt_dec && (bus_release_cnt != '0)) |-> bus_release_cnt == (bus_release_cnt -
       1'b1) ;
15  endproperty
```

Human Expert:

```
1  property bus_release_cnt_change_reset;
2  !rst_ni |-> !(|bus_release_cnt);
3   endproperty
4  property bus_release_cnt_change;
5  rst_ni && monitor_enable && !monitor_enable_q |-> multi_controller_enable_i |->
       bus_release_cnt == 30'(bus_inactive_timeout_i);
6   endproperty
7  property bus_release_cnt_change_load;
8  rst_ni && bus_release_cnt_load |-> bus_release_cnt == bus_release_cnt_sel;
9   endproperty
10 property bus_release_cnt_change_dec;
11  rst_ni && bus_release_cnt_dec && (|bus_release_cnt) |-> bus_release_cnt == bus_release_cnt -
       1;
12  endproperty
```

**Example 2**

Source Code:

```
1      if (state_sel_o           != mr_state_sel[i]       ||
2      add_rk_sel_o   != mr_add_rk_sel[i]   ||
3      key_full_sel_o  != mr_key_full_sel[i]  ||
4      key_dec_sel_o   != mr_key_dec_sel[i]   ||
5      key_words_sel_o != mr_key_words_sel[i] ||
```

```
6     round_key_sel_o != mr_round_key_sel[i]) begin
7     mr_err = 1'b1;
8     end
```

VERT Llama 3 Response:

```
1 property SyncErreotid;
2   (state_sel_o != mr_state_sel[i]      ||
3   add_rk_sel_o   != mr_add_rk_sel[i]  ||
4   key_full_sel_o != mr_key_full_sel[i]  ||
5   key_dec_sel_o  != mr_key_dec_sel[i]   ||
6   key_words_sel_o!= mr_key_words_sel[i] ||
7   round_key_sel_o!= mr_round_key_sel[i]) |-> mr_err = 1'b1;
8   endproperty
```

VERT DeepSeek Response:

```
1 property SyncErreotid;
2   (state_sel_o    != mr_state_sel[i]) ||
3   (add_rk_sel_o              != mr_add_rk_sel[i]) ||
4   (key_full_sel_o  != mr_key_full_sel[i]) ||
5   (key_dec_sel_o    != mr_key_dec_sel[i]) ||
6   (key_words_sel_o != mr_key_words_sel[i]) ||
7   (round_key_sel_o != mr_round_key_sel[i]) |-> mr_err == 1'b1 ;
8   endproperty
```

Human Expert:

```
1 property state_sel_cond;
2   (state_sel_o != mr_state_sel[i]) || (add_rk_sel_o != mr_add_rk_sel[i]) || (key_full_sel_o !=
      mr_key_full_sel[i]) || (key_dec_sel_o != mr_key_dec_sel[i]) || (key_words_sel_o !=
      mr_key_words_sel[i]) || (round_key_sel_o != mr_round_key_sel[i]) |-> mr_err;
3   endproperty
```

## A.6 ABLATION STUDY WITH UNCLEANED VARIABLE NAMES

Table 3 an ablation study with uncleaned variable names, presented below, to highlight the impact of cleaning variable names. The Table is split into two halves, containing Syntactically Correct Assertions and Functionally Correct Assertions. Columns 3 and 8 refer to the assertions generated by the base model. Columns 4 and 9 refer to assertions generated by a model fine-tuned on a dataset that contains syntactically incorrect variables (which refer to special characters not allowed in HDL languages); columns 5 and 10 refer to duplicate variables that may skew the model's learning and introduce ambiguity, and columns 6 and 11 refer to inconsistent variables such as conflicting variable names. Finally, columns 7 and 12 refer to the cleaned variables we eventually use to build VERT. The results provided in the table below demonstrate that failing to address syntactically incorrect variable names was the most critical, leading to lower performance in fine-tuned LLMs compared to even the base models. This is because the fine-tuned LLMs generate syntactically incorrect assertions stemming from erroneous variable names.

Table 3: Ablation Study with Uncleaned Variable Names

| Models | Benchmark/ Hardware IP | Syntactically Correct Assertions (%) | | | | | Functionally Correct Assertions (%) | | | | |
| --- | --- | --- | --- | --- | --- | --- | --- | --- | --- | --- | --- |
| | | Base Model | With Syntactically Incorrect Variables | With Duplicate Variables | With Inconsistent Variables | Cleaned Variables | Base Model | With Syntactically Incorrect Variables | With Duplicate Variables | With Inconsistent Variables | Cleaned Variables |
| Llama 3.1 | OpenTitan/AES | 35.84 | 35.20 | 72.80 | 86.40 | 88.70 | 8.02 | 7.20 | 68.80 | 82.40 | 83.48 |
| | OpenTitan/I2C | 29.53 | 28.57 | 66.67 | 83.33 | 83.33 | 9.39 | 9.52 | 66.67 | 80.16 | 83.33 |
| | OpenTitan/LC CTRL | 23.07 | 21.05 | 73.68 | 84.21 | 89.47 | 7.69 | 5.26 | 73.68 | 84.21 | 89.47 |
| | OpenTitan/ADC CTRL | 17.46 | 18.75 | 81.25 | 90.63 | 100.00 | 9.52 | 9.38 | 81.25 | 96.88 | 100.00 |
| | CVA6/Frontend | 41.18 | 38.46 | 76.92 | 92.31 | 92.31 | 11.76 | 15.38 | 76.92 | 84.62 | 92.31 |
| | CVA6/Decode&Issue | 22.58 | 23.53 | 82.35 | 94.12 | 100.00 | 6.45 | 5.88 | 82.35 | 97.06 | 100.00 |
| | CVA6/Execute | 25.55 | 24.76 | 74.29 | 85.71 | 91.43 | 5.45 | 5.71 | 74.29 | 86.67 | 91.43 |
| | CVA6/Commit | 38.57 | 35.44 | 73.42 | 88.61 | 89.87 | 10.00 | 10.13 | 73.42 | 89.87 | 89.87 |
| | CVA6/Controller&Top | 34.24 | 32.35 | 79.41 | 95.59 | 95.59 | 5.48 | 5.88 | 79.41 | 89.71 | 95.59 |
| | Pulpissimo/APB | 53.33 | 52.63 | 73.68 | 89.47 | 89.47 | 53.33 | 52.63 | 73.68 | 89.47 | 89.47 |
| | Pulpissimo/RISCV | 21.05 | 20.00 | 80.00 | 93.33 | 93.33 | 21.05 | 20.00 | 73.33 | 86.67 | 93.33 |
| | Pulpissimo/debug_unit | 16.67 | 18.18 | 81.82 | 90.91 | 100.00 | 16.67 | 18.18 | 81.82 | 90.91 | 100.00 |
| | **Average** | **29.92** | **29.08** | **76.36** | **89.55** | **92.79** | **13.73** | **13.76** | **75.47** | **88.22** | **92.36** |
| DeepSeek Coder | OpenTitan/AES | 10.81 | 10.19 | 75.80 | 89.17 | 94.90 | 6.08 | 6.37 | 75.80 | 92.36 | 93.63 |
| | OpenTitan/I2C | 12.12 | 11.29 | 79.03 | 95.16 | 97.58 | 8.33 | 8.06 | 79.03 | 95.16 | 97.58 |
| | OpenTitan/LC CTRL | 14.25 | 15.79 | 84.21 | 94.74 | 100.00 | 9.52 | 10.53 | 84.21 | 94.74 | 100.00 |
| | OpenTitan/ADC CTRL | 6.25 | 6.25 | 78.13 | 96.88 | 100.00 | 0.00 | 0.00 | 78.13 | 93.75 | 96.88 |
| | CVA6/Frontend | 56.25 | 57.14 | 78.57 | 92.86 | 92.86 | 37.50 | 35.71 | 78.57 | 85.71 | 92.86 |
| | CVA6/Decode&Issue | 18.92 | 18.75 | 84.38 | 96.88 | 100.00 | 13.51 | 12.50 | 84.38 | 93.75 | 100.00 |
| | CVA6/Execute | 26.37 | 25.25 | 77.78 | 97.98 | 97.98 | 20.88 | 20.20 | 77.78 | 92.93 | 97.98 |
| | CVA6/Commit | 21.65 | 20.43 | 75.27 | 82.80 | 89.25 | 17.53 | 17.20 | 75.27 | 81.72 | 89.25 |
| | CVA6/Controller&Top | 21.95 | 21.05 | 75.00 | 86.84 | 89.47 | 15.85 | 14.47 | 75.00 | 85.53 | 89.47 |
| | Pulpissimo/APB | 24 | 26.32 | 78.95 | 94.74 | 100.00 | 24.00 | 26.32 | 78.95 | 100.00 | 100.00 |
| | Pulpissimo/RISCV | 23.08 | 20.00 | 80.00 | 100.00 | 100.00 | 23.08 | 20.00 | 80.00 | 93.33 | 100.00 |
| | Pulpissimo/debug_unit | 15.38 | 18.18 | 81.82 | 100.00 | 100.00 | 15.38 | 18.18 | 81.82 | 90.91 | 100.00 |
| | **Average** | **20.92** | **20.89** | **79.08** | **94.00** | **96.84** | **15.97** | **15.80** | **79.08** | **91.66** | **96.47** |

Subsequently, LLMs fine-tuned with the cleaned variable list (VERT) performed up to 17.76% and 17.39% in syntactical and functional correctness, respectively, compared to the LLMs fine-tuned on a dataset that preserved duplicate variable names. This improvement is attributed to eliminating duplicates within the same assertion, which likely reduced ambiguity and enhanced the fine-tuned LLMs' ability to generate accurate results. Finally, addressing inconsistent variable names resulted in the smallest observed change, with the LLMs fine-tuned on the cleaned variable list increasing by up to 3.24% and 4.81%, in syntactical and functional correctness, respectively, compared to the LLMs fine-tuned on a dataset that maintained inconsistent variable names. This outcome is likely because such inconsistencies comprised a relatively minor portion of the overall variable list compared to the duplicates and syntactically incorrect variables.

A.7    EVALUATING IMPACT OF CONTAMINATION ON ASSERTION GENERATION

Regarding potential data contamination, from our experiments, we observed that certain HDL components negatively impact the generation of syntactically and functionally correct assertions. Notably, module instantiations and "ifdef" commands challenge assertion generation. Specifically, the models used to evaluate VERT rarely generate assertions from module instantiations, leading to syntactically and functionally incorrect results. Moreover, these smaller models tend to misinterpret "ifdef" commands as conventional if-else statements. While this misclassification occurs infrequently, it reduces the percentage of correctly generated assertions. Assertions derived from these commands are often both syntactically incorrect—since "ifdef" commands do not adhere to standard if-else syntax and lack the necessary information for typical branching—and functionally incorrect, as they do not contribute meaningfully to functional branching. In contrast, GPT-4o appears unaffected by these HDL components in its assertion generation. Table 1 illustrates the effect of increasing contamination in design files on assertion generation. Here, "contamination" refers to adding "ifdef" commands and module instantiations. For example, "+10 contamination" indicates that 10 additional instances of each of these elements (on top of the already existing instances) were introduced into the design files. The results show that as the level of contamination increases, the number of incorrectly generated assertions also rises. It should be noted that typically, in the hardware design, the number of "ifdef" commands is limited to at most five. Therefore, the scenarios used here to evaluate the contamination effect are unrealistic and to study the effect of contamination on the models.

Table 4 illustrates the effect of increasing contamination in design files on assertion generation. Column 1, "Models," specifies the LLM being tested, while Column 2, "Benchmark/Hardware IP," lists the specific test benchmark used. Columns 3, 4, and 5, under "Generated Assertions," indicate the total number of assertions generated with no contamination, an additional 10 contamination, and an additional 20 contamination in the input dataset, respectively. Columns 6, 7, and 8, under "Syntactically Correct (%)," measure the percentage of assertions that are syntactically valid for each contamination level. Finally, Columns 9, 10, and 11, under "Functionally Correct (%)," represent

Table 4: Effect of increasing contamination in design files on assertion generation

| Models | Benchmark/Hardware IP | Generated Assertions | | | Syntactically Correct (%) | | | Functionally Correct (%) | | |
|---|---|---|---|---|---|---|---|---|---|---|
| | | No Contamination | +10 Contamination | +20 Contamination | No Contamination | +10 Contamination | +20 Contamination | No Contamination | +10 Contamination | +20 Contamination |
| Llama 3.1 | OpenTitan/AES | 125 | 129 | 132 | 0.89 | 0.86 | 0.84 | 0.83 | 0.81 | 0.79 |
| | OpenTitan/I2C | 126 | 130 | 132 | 0.83 | 0.81 | 0.8 | 0.83 | 0.81 | 0.8 |
| | OpenTitan/LC CTRL | 19 | 21 | 22 | 0.89 | 0.81 | 0.77 | 0.89 | 0.81 | 0.77 |
| | OpenTitan/ADC CTRL | 32 | 34 | 36 | 1.00 | 0.94 | 0.89 | 1.00 | 0.94 | 0.89 |
| | CVA6/Frontend | 13 | 14 | 17 | 0.92 | 0.86 | 0.71 | 0.92 | 0.86 | 0.71 |
| | CVA6/Decode&Issue | 34 | 37 | 39 | 1.00 | 0.92 | 0.87 | 1.00 | 0.92 | 0.87 |
| | CVA6/Execute | 105 | 109 | 111 | 0.91 | 0.88 | 0.86 | 0.91 | 0.88 | 0.86 |
| | CVA6/Commit | 79 | 82 | 84 | 0.90 | 0.87 | 0.85 | 0.90 | 0.87 | 0.85 |
| | CVA6/Controller&Top | 68 | 71 | 73 | 0.96 | 0.92 | 0.89 | 0.96 | 0.92 | 0.89 |
| | Pulpissimo/APB | 19 | 21 | 23 | 0.89 | 0.81 | 0.74 | 0.89 | 0.81 | 0.74 |
| | Pulpissimo/RISCV | 15 | 17 | 18 | 0.93 | 0.82 | 0.78 | 0.93 | 0.82 | 0.78 |
| | Pulpissimo/debug_unit | 11 | 14 | 14 | 1.00 | 0.79 | 0.79 | 1.00 | 0.79 | 0.79 |
| DeepSeek Coder | OpenTitan/AES | 157 | 161 | 164 | 0.95 | 0.93 | 0.91 | 0.94 | 0.91 | 0.9 |
| | OpenTitan/I2C | 124 | 129 | 131 | 0.98 | 0.94 | 0.92 | 0.98 | 0.94 | 0.92 |
| | OpenTitan/LC CTRL | 19 | 22 | 23 | 1.00 | 0.86 | 0.83 | 1.00 | 0.86 | 0.83 |
| | OpenTitan/ADC CTRL | 32 | 35 | 35 | 1.00 | 0.91 | 0.91 | 0.97 | 0.89 | 0.89 |
| | CVA6/Frontend | 14 | 16 | 18 | 0.93 | 0.81 | 0.72 | 0.93 | 0.81 | 0.72 |
| | CVA6/Decode&Issue | 32 | 34 | 35 | 1.00 | 0.94 | 0.91 | 1.00 | 0.94 | 0.91 |
| | CVA6/Execute | 99 | 102 | 104 | 0.98 | 0.95 | 0.93 | 0.98 | 0.95 | 0.93 |
| | CVA6/Commit | 93 | 95 | 96 | 0.89 | 0.87 | 0.86 | 0.89 | 0.87 | 0.86 |
| | CVA6/Controller&Top | 76 | 81 | 81 | 0.89 | 0.84 | 0.84 | 0.89 | 0.84 | 0.84 |
| | Pulpissimo/APB | 19 | 22 | 22 | 1.00 | 0.86 | 0.86 | 1.00 | 0.86 | 0.86 |
| | Pulpissimo/RISCV | 15 | 18 | 19 | 1.00 | 0.83 | 0.79 | 1.00 | 0.83 | 0.79 |
| | Pulpissimo/debug_unit | 11 | 13 | 15 | 1.00 | 0.85 | 0.73 | 1.00 | 0.85 | 0.73 |

the percentage of assertions that are logically accurate and align with the intended functionality under the same contamination conditions. Here, "contamination" refers to the addition of "ifdef" commands and module instantiations. A contamination level lower than 10 was found to have a negligible impact on the results, while levels exceeding 20 were impractical due to exceeding the context size limitations of our models. The results show that as the level of contamination increases, the number of incorrectly generated assertions also rises. This results in a 3% drop in accuracy. It should be noted that typically, in the hardware design, the number of "ifdef" commands is limited to at most five. Therefore, the scenarios used here to evaluate the contamination effect are unrealistic and for the purpose of studying the effect of contamination on the models.

### A.8 Assessing Functional Correctness and Relevance of Assertions through Mutation Testing

The functional correctness of the generated assertions, as presented in Table 1, was evaluated using mutation testing, consistent with the methodology outlined in [1]. This approach involved introducing intentional, small code modifications (mutants) that deviated from the expected assertion logic. By detecting these mutants, we demonstrated the effectiveness of the generated assertions in identifying logical inconsistencies and validating their utility. Unlike trivial or redundant assertions (e.g., "assert True"), these assertions were intricately aligned with the critical components of the hardware design, ensuring their relevance and impact. Our findings revealed that the generated assertions achieved up to 100% functional correctness or ability to detect mutations across various benchmarks, underscoring their robustness and effectiveness. The same mutation testing methodology from [1] was also applied to verify the importance of the generated assertions, further affirming their significance. This comprehensive evaluation highlighted the non-redundant nature of the benchmarks and the potential of LLMs fine-tuned on VERT for hardware verification. Furthermore, the method ensured complete coverage of conditional branches and critical logic paths within the hardware design. The LLM-generated assertions were specifically crafted to validate every logical path, leaving no branch or condition unchecked, thereby reinforcing their role in achieving thorough hardware verification.

### A.9 Coverage Measurement

Complete Path Coverage (CPC) refers to covering all possible independent paths within an automaton. A path begins at the initial node, traverses through the graph's edges, and ends at a final node. While CPC provides a thorough examination of the system, it becomes infeasible for graphs containing cycles, as these can result in infinite path lengths.

In our approach, we use complete path coverage as our primary coverage metric. This ensures a comprehensive evaluation of the system's behavior by accounting for all potential paths. To validate our coverage, we employed both formal and simulation-based verification tools, including Cadence JasperGold and Xilinx Vivado. These tools allowed us to rigorously analyze the generated assertions and ensure that they comprehensively cover all the functions defined within the system.

By leveraging our method to extract properties from every possible conditional branch, we achieve up to 100% coverage. This robust verification strategy confirms the correctness and reliability of the automaton's functionality across all defined behaviors.

