# OpenReview forum: "VERT: A SystemVerilog Assertion Dataset to Improve Hardware Verification with LLMs"
_ICLR.cc/2025/Conference — Submitted to ICLR 2025_

### Official Review · Reviewer_LBj3 · 2024-11-03

**Soundness:** 2
**Presentation:** 3
**Contribution:** 3
**Rating:** 5
**Confidence:** 3

**Summary:**

This paper proposes an open-source dataset named VERT to improve SystemVerilog assertion generation for large language models (LLMs). The authors identify existing gaps in the verification performance of off-the-shelf LLMs through examples. Their evaluation shows that VERT-fine-tuned LLMs achieve significant improvements in functional correctness and syntactical coverage.

**Strengths:**

- Timely problem
- Good solution

**Weaknesses:**

- Insufficient evaluation on verification effectiveness

**Questions:**

Thank you for submitting to ICLR 2025. The research problem is both interesting and timely. The motivating examples are helpful for me to understand existing problems in the off-the-shelf models, and the experimental results show significant improvements.

The authors evaluated the effectiveness of their approach based on the total number of assertions and correctness in both syntax and functionality. However, these metrics do not reflect how important their assertions are. For example, one can generate a working assertion like 'assert True', but it has no impact on functional verification. It would be helpful if the authors could compare the importance of assertions, for example, using mutation tests: https://ieeexplore.ieee.org/abstract/document/10546742

---

> ### Author Response · Authors · 2024-11-21
> **Response to Reviewer LBj3**
>
> We thank the reviewer for their thoughtful and insightful comment regarding the evaluation metrics for our approach. Below, we address the concerns and provide clarifications:
>
> The functional correctness of the generated assertions, as presented in Table 1 in the manuscript, was evaluated using mutation testing, consistent with the methodology outlined in [1]. This approach involved introducing intentional, small code modifications (mutants) that deviated from the expected assertion logic. By detecting these mutants, we demonstrated the effectiveness of the generated assertions in identifying logical inconsistencies and validating their utility. Unlike trivial or redundant assertions (e.g., "assert True"), these assertions were intricately aligned with the critical components of the hardware design, **ensuring their relevance and impact**.
>
> Our findings revealed that the generated assertions achieved up to 100% functional correctness or ability to detect mutations across various benchmarks, underscoring their robustness and effectiveness. The same mutation testing methodology from [1] was also applied to verify the importance of the generated assertions, further affirming their significance. This comprehensive evaluation highlighted the non-redundant nature of the benchmarks and the potential of LLMs fine-tuned on VERT for hardware verification. Furthermore, the method ensured complete coverage of conditional branches and critical logic paths within the hardware design. The LLM-generated assertions were specifically crafted to validate every logical path, leaving no branch or condition unchecked, thereby reinforcing their role in achieving thorough hardware verification. We have incorporated this into our manuscript (highlighted in blue) in Section 5.1 and further elaborated in Appendix A.8.
>
>
> [1] M. Iman et al., ARTmine: Automatic Association Rule Mining with Temporal Behavior for Hardware Verification: DATE 2024.

---

> ### Author Response · Authors · 2024-11-27
>
> Dear Reviewer LBj3,
>
> Thank you once again for your insightful comments and feedback on our paper. We have carefully considered your suggestions and prepared a detailed rebuttal in response. We sincerely appreciate reviewer Mh3q for reconsidering their score in light of our response. We would be happy to have your feedback as well and are grateful for any further thoughts you may have. We look forward to your thoughts on our rebuttal.

---

### Official Review · Reviewer_piHx · 2024-11-03

**Soundness:** 2
**Presentation:** 3
**Contribution:** 2
**Rating:** 5
**Confidence:** 4

**Summary:**

The article introduces a new dataset named VERT, specifically designed for generating SystemVerilog assertions to enhance the efficiency of the hardware verification process. Hardware verification is crucial in modern chip design, accounting for 70% of development time. However, the current reliance on manual assertion generation is inefficient and prone to errors. VERT systematically collects and augments variables from open-source hardware description languages, generating synthetic code snippets and corresponding assertions. This dataset aims to optimize the performance of smaller open-source large language models (such as DeepSeek Coder and Llama 3.1), enabling them to surpass proprietary models like GPT-4o. Additionally, the VERT dataset supports local fine-tuning, eliminating costly licensing fees and ensuring data privacy, thus providing a scalable and cost-effective solution for automated hardware verification.

**Strengths:**

1. The VERT dataset provides a high-quality data source for generating SystemVerilog assertions, helping small open-source large language models generate accurate assertions in hardware verification.

2. With fine-tuning on the VERT dataset, smaller open-source models (such as DeepSeek Coder and Llama 3.1) can surpass proprietary models like GPT-4o in generating accurate assertions, demonstrating the feasibility and efficiency of open-source solutions in hardware verification.

3.  As an open-source dataset, VERT provides researchers and companies with the flexibility to expand it with additional hardware design scenarios as needed, laying a solid foundation for future research and applications in hardware verification.

**Weaknesses:**

1. The authors compared the test results of GPT-4o and other models fine-tuned on VERT in Figure 4. Did the authors compare VERT with current benchmarks and remove data that could potentially contaminate the test set?

2. I would recommend that the authors consider submitting this article to a hardware-focused conference or journal, as the primary contribution of the work lies in the dataset generation process for VERT rather than in the field of artificial intelligence or deep learning. As such, I feel it somewhat diverges from the main objectives of ICLR:

> The International Conference on Learning Representations (ICLR) is the premier gathering of professionals dedicated to the advancement of the branch of **artificial intelligence** called representation learning, but generally referred to as **deep learning**.

**Questions:**

See Weakness.

---

> ### Author Response · Authors · 2024-11-21
> **Response to Reviewer piHx**
>
> 1.  We thank the reviewer for raising this important point regarding comparison with other benchmarks. To the best of our knowledge, there are currently no publicly available benchmarks specifically designed for SystemVerilog assertion generation in hardware verification. Existing datasets and benchmarks focus primarily on Verilog code generation or hardware design, which makes direct comparisons with other datasets infeasible.
>
>     Regarding potential data contamination, from our experiments, we observed that certain HDL components could negatively impact the generation of syntactically and functionally correct assertions. Notably, module instantiations and `ifdef` commands pose challenges for assertion generation. Specifically, the models used to evaluate VERT rarely generate assertions from module instantiations, leading to syntactically and functionally incorrect results.
>
>     Moreover, these smaller models tend to misinterpret `ifdef` commands as conventional if-else statements. While this misclassification occurs infrequently, it reduces the percentage of correctly generated assertions. Assertions derived from these commands are often both syntactically incorrect—since `ifdef` commands do not adhere to standard if-else syntax and lack the necessary information for typical branching—and functionally incorrect, as they do not contribute meaningfully to functional branching. In contrast, GPT-4o appears to be unaffected by these HDL components in its assertion generation.
>
>     Table 1 shows the impact of increasing contamination in design files on assertion generation. Column 1 lists the tested LLMs. Columns 2–4 ("Syntactically Correct (%)") display the percentage of syntactically valid assertions under no contaminants, an additional 10 contaminants, and an additional 20 contaminants. Columns 6–8 ("Functionally Correct (%)") show the percentage of assertions that are logically accurate and functionally aligned under the same contamination conditions.
>     Here, “contamination” refers to the addition of `ifdef` commands and module instantiations. A contamination level lower than 10 was found to have a negligible impact on the results, while levels exceeding 20 were impractical due to exceeding the context size limitations of our models. The results show that as the level of contamination increases, the number of incorrectly generated assertions also rises. This results in a 3% drop in accuracy.  **It should be noted that typically, in the hardware design, the number of `ifdef` commands is limited to at most five. Therefore, the scenarios used here to evaluate the contamination effect are unrealistic and for the purpose of studying the effect of contamination on the models.** We have incorporated more detailed results into Appendix A.7 of our manuscript.
>
>
>     | Models |  | Syntactically Correct (%) |  |  | Functionally Correct (%)  |  |
>     |---|---|---|---|---|---|---|
>     |  | No Contamination | +10 Contamination | +20 Contamination | No Contamination | +10 Contamination | +20 Contamination |
>     | Llama 3.1 | 0.92 | 0.85 | 0.82 | 0.92 | 0.85 | 0.81 |
>     | DeepSeek | 0.96 | 0.88 | 0.85 | 0.96 | 0.88 | 0.85 |
>
>
> 2. While the popular focus areas of the ICLR conference revolve around advancements in artificial intelligence and deep learning, the growing use of AI for automation in diverse domains, particularly in electronic design automation (EDA), is creating new and exciting application-centric tracks.
>
>     We believe that while hardware-focused venues are a viable option, topics such as novel automation through dataset generation and LLM-driven EDA pipelines, including design verification, are highly relevant to ICLR. Similar emerging areas have been showcased at ICLR in the past, gaining traction and recognition. For instance, *Hw-nas-bench: Hardware-aware neural architecture search benchmark* [1], a hardware-centric benchmark paper, was presented during the rise of NAS, and *Retrieval-Guided Reinforcement Learning for Boolean Circuit Minimization* [2] was featured at ICLR 2024. These examples highlight how ICLR often welcomes emerging interdisciplinary communities under appropriate themes. We kindly request the reviewer to evaluate this draft under the relevant themes outlined in the ICLR CFP.
>
>     1.	Infrastructure, software libraries, hardware, etc.
>     2.	Applications to robotics, autonomy, planning
>     3.	Datasets and benchmarks
>
>     Our work, which focuses on the novel automation of hardware design verification through dataset generation and LLMs, fits within these themes and represents an important and emerging topic in the field.
>     We thank the reviewer for their feedback and hope these clarifications address their concerns.
>
>
> [1] C. Li et al., HW-NAS-Bench: Hardware-Aware Neural Architecture Search Benchmark. ICLR 2024
>
> [2] A. Chowdhury et al., Retrieval-Guided Reinforcement Learning for Boolean Circuit Minimization. ICLR 2024

---

> ### Author Response · Authors · 2024-11-27
>
> Dear Reviewer piHx,
>
> Thank you once again for your insightful comments and feedback on our paper. We have carefully considered your suggestions and prepared a detailed rebuttal in response. We sincerely appreciate reviewer Mh3q for reconsidering their score in light of our response. We would be happy to have your feedback as well and are grateful for any further thoughts you may have. We look forward to your thoughts on our rebuttal.

---

### Official Review · Reviewer_Mh3q · 2024-11-04

**Soundness:** 3
**Presentation:** 3
**Contribution:** 2
**Rating:** 6
**Confidence:** 4

**Summary:**

This paper mainly proposes VERT, a dataset specially designed for improving the generation of SystemVerilog assertions using LLMs in hardware verification. The dataset is generated synthetically and contains a rich variety of conditions based on the cleaned variable list, allowing the model to learn complex hardware logic and generate accurate assertions without oversimplifying or omitting critical conditions. Experimental results show that small LLMs fine-tuned using the VERT dataset significantly outperform base models and GPT-4o in terms of syntax and functional correctness.

**Strengths:**

1.	This paper is generally well-written and easy to follow, except some points. I can understand most statements easily. For some improvement suggestions, please see the weakness part below.
2.	The proposed dataset enables the small LLMs to significantly improve performance compared to the base and surpass GPT-4o without fine-tuning. And it also shows good effectiveness in real task scenarios.

**Weaknesses:**

1.	Regarding the metrics of compilation success and simulation performance mentioned in the main contribution (4), I did not find any description of this part in the experimental section of the paper.
2.	The paper’s experimental section is not comprehensive. It does not adequately validate the presented motivations and lacks some necessary ablation studies.
a)	Will the constructed dataset and the test dataset have the same modules, causing issues with training data leakage?
b)	In section 4.1, regarding the intuition in dataset formulation, did the assertions generated by LLMs successfully address these issues?
c)	One important assumption made during the dataset construction is that the reuse of IPs would make it difficult for the model to differentiate between components. I think that an ablation study using a dataset constructed with an uncleaned variable list is needed to demonstrate this point.
3.	What is the coverage of the assertions generated by large language models?"
4.	I would like to know how the assertions generated by the fine-tuned LLMs compared to those designed by industrial experts in real-world task scenarios and the synthetic assertions through the method in section 4.3?

**Questions:**

1.	Regarding the metrics of compilation success and simulation performance mentioned in the main contribution (4), I did not find any description of this part in the experimental section of the paper.
2.	The paper’s experimental section is not comprehensive. It does not adequately validate the presented motivations and lacks some necessary ablation studies.
a)	Will the constructed dataset and the test dataset have the same modules, causing issues with training data leakage?
b)	In section 4.1, regarding the intuition in dataset formulation, did the assertions generated by LLMs successfully address these issues?
c)	One important assumption made during the dataset construction is that the reuse of IPs would make it difficult for the model to differentiate between components. I think that an ablation study using a dataset constructed with an uncleaned variable list is needed to demonstrate this point.
3.	What is the coverage of the assertions generated by large language models?"
4.	I would like to know how the assertions generated by the fine-tuned LLMs compared to those designed by industrial experts in real-world task scenarios and the synthetic assertions through the method in section 4.3?

---

> ### Author Response · Authors · 2024-11-21
> **Response to Reviewer Mh3q (Part 1/4)**
>
> 1) **Metrics for Compilation Success and Simulation Performance:** We thank the reviewer for raising the need for more clarity in our terminology. To clarify, **Compilation Success** refers to the compliance of the generated assertions with hardware description language standards, indicating they are **Syntactically correct**. In contrast, **Simulation Performance** assesses whether the assertions accurately reflect the intended hardware behavior, validating their expected functionality and confirming that they are **Functionally correct**.
>
>     We acknowledge that this distinction was not sufficiently articulated in the original contributions of the manuscript. To address this, we have revised our writing to better align with the terminology used throughout the manuscript.  In specific, the updated contribution now reads: “Additionally, we evaluate the fine-tuned models using metrics such as syntactic and functional correctness, achieving up to 100% accuracy on modules from industry-standard SoCs, including OpenTitan, Pulpissimo and CVA6, thereby demonstrating the reliability and effectiveness of VERT in real-world scenarios.”
>
>     Furthermore, the terms **Syntactically correct** and **Functionally correct** have been consistently applied in **Table 1** and **Section 5.2** of the manuscript to evaluate and report the performance of the fine-tuned models. This ensures alignment with the metrics defined throughout the manuscript, reinforcing the rigor and transparency of our evaluation methodology.
>
> 2)  a. **Training data leakage:** We ensure that there is no overlap between the constructed dataset and the test benchmarks. Specifically, assertion examples from OpenTitan, CVA6, and Pulpissimo are not used in generating VERT.
> As detailed in **Figure 1** of the manuscript, the **constructed dataset** (VERT) is synthesized from a curated and cleaned list of variables extracted from various open-source projects, including BOOM-core, Rocket-chip, and XiangShan. This figure provides a clear overview of the data sources and the systematic methodology employed, ensuring transparency and demonstrating the diversity and representativeness of the dataset for assertion generation.
>    The **test benchmark**, on the other hand, consists of source code specifically derived from real-world System-on-Chip (SoC) designs, such as OpenTitan, CVA6, and Pulpissimo. These are distinct from the constructed dataset as they use actual hardware module code rather than synthetically generated code.

---

> ### Author Response · Authors · 2024-11-21
> **Response to Reviewer Mh3q (Part 2/4)**
>
> 2) b. **VERT Addressing Issues of existing LLM-based generated assertions:** As outlined in the manuscript, currently, GPT-4o faces several key challenges, including (1) clock cycle and pre-condition capturing, (2) Miscapturing if Condition for else Branches, (3) Nested If-Else Conditions, and (4) handling long conditions.  In this response, we demonstrate, using examples as follows, how LLMs trained on VERT address these issues. Relevant examples pertaining to this response have been included in Appendix A.5.1 of the manuscript.
>
>     1. **Clock Cycle Misinterpretations:** We observe that LLMs often get confused between using the overlapping implication symbol ($|->$) and the non-overlapping symbol ($|=>$). To resolve clock cycle misinterpretation by LLMs, we standardized our format by using the overlapping implication symbol with a specified delay count and replacing the non-overlapping symbol. As shown in Text Box A.1 in Appendix A.5.1 of our manuscript, GPT-4o incorrectly uses the non-overlapping symbol, but both LLMs fine-tuned on VERT address this issue by correctly interpreting the clock cycle by using the overlapping implication symbol.
>     2. **Miscapturing if Condition for else Branches:** LLMs often overlook the previous if/else if conditions when generating assertions for the subsequent else or else if branches in conditional statements. VERT addresses the common omission of conditions in the else/else-if branches of if-else statements by exposing the model to diverse conditional structures.  As shown in Text Box A.2 in Appendix A.5.1 of our manuscript,  GPT-4o, in its second assertion, misses that the second assertion should verify that `monitor_enable` is `FALSE` and `monitor_enable_q` is `TRUE`. However, both LLMs fine-tuned on VERT correctly identify these conditions and include them in the fourth generated assertion.
>     3.	**Nested If-Else Conditions:** LLMs often struggle with generating assertions for deeply nested if-else conditions (i.e., blocks nested beyond two levels), which are common in hardware design. To address the challenge of LLMs struggling with deeply nested if-else statements, we expanded VERT to include complex, multi-level conditional structures. As shown in Text Box A.2 in Appendix A.5.1 of our manuscript, GPT-4o incorrectly represents the logic of a nested block. Specifically, it ignores the if condition and creates an assertion using a ternary operator, where the inner if condition is used as the test, the assignment is the `TRUE` branch, and `1'b1` is the false branch. However, both LLMs fine-tuned on VERT to correctly identify the correct nested statement and their conditions.
>     4.  **Handling Long Conditions:** To address the challenge of generating accurate assertions for long and complex conditions, we expanded VERT to include a variety of cases where multiple conditions and operators must be evaluated simultaneously. As shown in Text Box A.3 in Appendix A.5.1 of our manuscript, the source code requires any one of six conditions to be met before raising the `mr_err` flag. However, the generated output by GPT- 4o creates three assertions for one flag change, each neglecting the other three conditions. This not only leads to incorrect assertion generation but also causes overhead in the number of assertions. However, both LLMs fine-tuned on VERT correctly generate a singular assertion that describes all the conditions and their relationships within the if-statement.

---

> ### Author Response · Authors · 2024-11-21
> **Response to Reviewer Mh3q (Part 3/4)**
>
> 2) c. **Ablation Study with Uncleaned Variable Names:** We appreciate the suggestion and have conducted an ablation study with uncleaned variable names, presented in the Table below, to highlight the impact of cleaning variable names. The Table is split into two halves, containing the average percentage of Syntactically Correct Assertions and Functionally Correct Assertions across all the benchmarks outlined in Section 5.1. Columns 2 and 7 refer to the assertions generated by the base model. Columns 3 and 8 refer to assertions generated by a model fine-tuned on a dataset that contains syntactically incorrect variables (which refer to special characters not allowed in HDL languages); columns 4 and 9 refer to duplicate variables that may skew the model’s learning and introduce ambiguity, and columns 5 and 10 refer to inconsistent variables such as conflicting variable names. Finally, columns 6 and 11 refer to the cleaned variables we eventually use to build VERT. The results provided in the Table below demonstrate that failing to address syntactically incorrect variable names was the most critical, leading to lower performance in fine-tuned LLMs compared to even the base models. This is because the fine-tuned LLMs generate syntactically incorrect assertions stemming from erroneous variable names.
>
>     Subsequently, LLMs fine-tuned with the cleaned variable list (VERT) performed up to 17.76% and  17.39% in syntactical and functional correctness, respectively, compared to the LLMs fine-tuned on a dataset that preserved duplicate variable names. This improvement is attributed to eliminating duplicates within the same assertion, which likely reduced ambiguity and enhanced the fine-tuned LLMs' ability to generate accurate results.
>
>     Finally, addressing inconsistent variable names resulted in the smallest observed change, with the LLMs fine-tuned on the cleaned variable list increasing by up to 3.24% and 4.81%, in syntactical and functional correctness, respectively, compared to the LLMs fine-tuned on a dataset that maintained inconsistent variable names. This outcome is likely because such inconsistencies comprised a relatively minor portion of the overall variable list compared to the duplicates and syntactically incorrect variables. We have provided more details in Table 3 and Appendix A.6 of our manuscript.
>
>
> | Models    | Syntactically Correct Assertions(%) |                                        |                          |                             |                   | Functionally Correct Assertions(%) |                                        |                          |                             |                   |
> |-----------|-------------------------------------|----------------------------------------|--------------------------|-----------------------------|-------------------|------------------------------------|----------------------------------------|--------------------------|-----------------------------|-------------------|
> |           | Base Model                          | With Syntactically Incorrect variables | With Duplicate Variables | With Inconsistent Variables | Cleaned Variables | Base Model                         | With Syntactically Incorrect variables | With Duplicate Variables | With Inconsistent Variables | Cleaned Variables |
> | LLama 3   | 29.92                               | 29.08                                  | 76.36                    | 89.55                       | 92.79             | 13.73                              | 13.76                                  | 75.47                    | 88.22                       | 92.36             |
> | DeepSeek  | 20.92                               | 20.89                                  | 79.08                    | 94                          | 96.84             | 15.97                              | 15.8                                   | 79.08                    | 91.66                       | 96.47             |

---

> ### Author Response · Authors · 2024-11-21
> **Response to Reviewer Mh3q (Part 4/4)**
>
> 3) **Coverage Statistic:** Complete Path Coverage (CPC) refers to covering all possible independent paths within an automaton. A path begins at the initial node, traverses through the graph's edges, and ends at a final node. While CPC provides a thorough examination of the system, it becomes infeasible for graphs containing cycles, as these can result in infinite path lengths [1].
> In our approach, we use complete path coverage as our primary coverage metric. This ensures a comprehensive evaluation of the system's behavior by accounting for all potential paths. To validate our coverage, we employed both formal and simulation-based verification tools, including **Cadence JasperGold** and **Xilinx Vivado**. These tools allowed us to analyze the generated assertions and ensure that they cover all the functions defined within the system. By leveraging our method to extract properties from every possible conditional branch, we achieve up to 100% coverage. The information has been included and highlighted in Section 5.1 of the manuscript.
>
> [1] J. Tong et al ., Defining and providing coverage for assertion-based dynamic verification. 2010 Journal of Electronic Testing 26
>
> 4) **Comparison of LLMs fine-tuned on VERT with human experts:** Human experts and LLMs fine-tuned on VERT generate assertions differently. Experts often create simpler, more readable assertions, reflecting a preference for human-friendly formats. LLMs, on the other hand, produce more standardized logical expressions. This difference highlights style preferences rather than superiority. **Notably, despite these stylistic differences, LLMs fine-tuned on VERT achieve assertion generation performance comparable to that of human experts.** We have provided examples in Text Box A.4 in Appendix A.5.2 of the manuscript. In Example 1, for instance, when verifying whether the bus_release_cnt signal has been reset, the LLM-generated assertion takes the form `bus_release_cnt == '0'`, while the human expert expresses the same operation as `!(|bus_release_cnt).` The latter format, often favored by experts, is typically chosen for its ease of writing and brevity.

---

> > ### Comment · Reviewer_Mh3q · 2024-11-26
> >
> > Thank you for your detailed responses. Most of my concerns have been addressed. I will increase my score to 6.

---

> > > ### Author Response · Authors · 2024-11-27
> > >
> > > Thank you very much for the prompt response and valuable feedback on our paper. We greatly appreciate the time and effort dedicated to the detailed review of our work, and the input provided has been instrumental in strengthening our submission.

---

### Comment · Area_Chair_KHNG · 2024-11-25

Dear Reviewers,

This is a kind reminder that the dicussion phase will be ending soon on November 26th. Please read the author responses and engage in a constructive discussion with the authors.

Thank you for your time and cooperation.

Best,

Area Chair

---

### Meta-Review · Area_Chair_KHNG · 2024-12-20

**Metareview:**

This paper introduces VERT, an open-source dataset designed for generating SystemVerilog assertions using large language models (LLMs) for hardware verification. The dataset is synthetically generated under various conditions, and experiments show that small LLMs fine-tuned on VERT outperform both base models and GPT-4 in terms of syntax and functional correctness.

However, two of the three reviewers have recommended rejection, raising several concerns. First, the authors fail to compare VERT against existing datasets or control for potential test set contamination. Second, the paper's focus on hardware dataset generation seems misaligned with the scope of the ICLR venue, suggesting it might be better suited for hardware-related conferences. Finally, the evaluation metrics used are inadequate, as they do not adequately capture the significance of the generated assertions or provide a detailed analysis of the mutation testing results.

In conclusion, I do not recommend accepting this paper in its current form.

**Additional Comments On Reviewer Discussion:**

Reviewers Mh3q, piHx, and LBj3 rated this paper as 6: borderline accept (kept the score), 5: borderline reject (kept the score), and 5: borderline reject (kept the score), respectively. The reviewers raised the following concerns:

- Limited insights (Reviewer LBj3)

- Scope misalignment (Reviewer piHx)

- Insufficient experiments (Reviewer LBj3)

- Unclear experiment details (Reviewers Mh3q, piHx)

In the rebuttal, the authors addressed some concerns by providing additional experiments and clarifying certain aspects of the explanation. However, several fundamental issues remain inadequately addressed. First, the paper lacks quantitative results or detailed analysis of the mutation testing. Second, the authors fail to compare VERT with existing datasets or control for potential test set contamination.

Therefore, I do not recommend accepting this paper in its current form.

---

### Decision · Program_Chairs · 2025-01-22

Reject